# GRAPH TRANSFORMERS DREAM OF ELECTRIC FLOW

**Xiang Cheng**[1]**, Lawrence Carin**[1]**, Suvrit Sra**[2]
[1]Department of Electrical and Computer Engineering, Duke University, USA
[2]School of Computation, Information and Technology, Technical University of Munich, Germany
`{xiang.cheng, lcarin}@duke.edu,   s.sra@tum.de`

## ABSTRACT

We show theoretically and empirically that the linear Transformer, when applied to graph data, can implement algorithms that solve canonical problems such as electric flow and eigenvector decomposition. The Transformer has access to information on the input graph only via the graph's incidence matrix. We present explicit weight configurations for implementing each algorithm, and we bound the constructed Transformers' errors by the errors of the underlying algorithms. Our theoretical findings are corroborated by experiments on synthetic data. Additionally, on a real-world molecular regression task, we observe that the linear Transformer is capable of learning a more effective positional encoding than the default one based on Laplacian eigenvectors. Our work is an initial step towards elucidating the inner-workings of the Transformer for graph data. Code is available at
https://github.com/chengxiang/LinearGraphTransformer

## 1 INTRODUCTION

The Transformer architecture (Vaswani et al., 2017) has seen great success in the design of graph neural networks (GNNs) (Dwivedi & Bresson, 2020; Ying et al., 2021; Kim et al., 2022; Müller et al., 2023), and it has demonstrated impressive performance in many applications ranging from prediction of chemical properties to analyzing social networks (Yun et al., 2019; Dwivedi et al., 2023; Rong et al., 2020). In contrast to this empirical success, the underlying reasons for *why* Transformers adapt well to graph problems are less understood. Several works study the *expressivity* of attention-based architectures (Kreuzer et al., 2021; Kim et al., 2021; 2022; Ma et al., 2023), but such analyses often rely on the ability of neural networks to approximate arbitrary functions, and may require a prohibitive number of parameters.

This paper is motivated by the need to understand, at a mechanistic level, how the Transformer processes graph-structured data. Specifically, we study the ability of a *linear Transformer* to solve certain classes of graph problems. The linear Transformer is similar to the standard Transformer, but softmax-based activation is replaced with linear attention, and MLP layers are replaced by linear layers. Notably, the linear Transformer contains no *a priori* knowledge of the graph structure; all information about the graph is provided via an *incidence matrix* $B$. For unweighted graphs, the columns of $B$ are just $\{-1, 0, 1\}$-valued indicator vectors that encode whether an edge touches a vertex; no other explicit positional or structural encodings are provided.

Even in this minimal setup, we are able to design *simple* configurations of the weight matrices of a Transformer that enable it to solve fundamental problems such as electric flow and Laplacian eigenvector decomposition. Furthermore, we provide explicit error bounds that scale well with the number of Transformer layers. Several of our constructions rely crucially on the structure of the linear attention module, and may help shed light on the success of attention-based GNNs. We hope that our analysis paves the way to a better understanding of the learning landscape of graph Transformers, such as concrete bounds on their generalization and optimization errors.

Besides enhancing our understanding of Transformers, our results are also useful for the practical design of graph Transformers. In Sections 3 and 4, we show that the same linear Transformer architecture is capable of learning a number of popular positional encodings (PE). In Section 6, we provide experimental evidence that the linear Transformer can learn better PEs than hard-coded PEs.

| Lemma | Task | Transformer Implements: | # layers for $\varepsilon$ error |
|---|---|---|---|
| 1 | Electric Flow ($\mathcal{L}^\dagger$) | Gradient Descent | $\log(1/\varepsilon)$ |
| 4 | Electric Flow ($\mathcal{L}^\dagger$) | Multiplicative Expansion | $\log\log(1/\varepsilon)$ |
| 2 | Resistive Embedding ($\sqrt{\mathcal{L}^\dagger}$) | Power Series | $\log(1/\varepsilon)$ |
| 3 | Heat Kernel ($e^{-s\mathcal{L}}$) | Power Series | $\log(1/\varepsilon) + s\lambda_{\max}$ |
| 5 | Heat Kernel ($e^{-s\mathcal{L}}$) | Multiplicative Expansion | $\log(1/\varepsilon) + \log(s\lambda_{\max})$ |
| 6 & 7 | $k$-Eigenvector Decomp. of $\mathcal{L}$ | Subspace Iteration | (subspace-iteration steps)$/k$ |

Table 1: Summary of Error Bounds proved for various Transformer constructions. $\mathcal{L}$ is the graph Laplacian; $\mathcal{L}^\dagger$ denotes its pseudoinverse.

## 1.1 SUMMARY OF CONTRIBUTIONS

Below, we summarize the main contributions of our paper.

1. We provide explicit weight configurations for which the Transformer ***implements efficient algorithms for several fundamental graph problems***. These problems serve as important primitives in various graph learning algorithms, and have also been useful as PEs in state-of-the-art GNNs.

   (a) Lemma 1 constructs a Transformer that solves electric flows by ***implementing steps of gradient descent to minimize flow energy***; consequently, it can also invert the graph Laplacian.

   (b) Lemmas 2 and 3 construct Transformers that compute low-dimensional resistive embeddings and heat kernels. Both constructions are based on implementing suitable power series.

   (c) By implementing a ***multiplicative polynomial expansion***, Lemma 4 provides a construction for electric flow with exponentially higher accuracy than Lemma 1. Similarly, Lemma 5 provides a construction that computes the heat kernel using much fewer layers than Lemma 3.

   (d) In Lemma 6, we show that the ***Transformer can implement subspace iteration for finding the top-$k$ (or bottom-$k$) eigenvectors of the graph Laplacian***. Central to this analysis is the ability of self-attention to compute a QR decomposition of the feature vectors.

   We derive **explicit error bounds** for the Transformer based on the **convergence rates of the underlying algorithm** implemented by the Transformer. We summarize these results in Table 1.

2. In Section 5, we propose a more efficient version of the linear Transformer with much fewer parameters. We show that this more efficient linear Transformer can nonetheless implement all the above-mentioned constructions. Further, we show in Lemma 10 that the parameter-efficient linear Transformer has desirable invariance and equivariance properties.

3. We test the empirical performance of our theory on synthetic random graphs. In Section 3.5, we verify that Transformers with a few layers can achieve high accuracy in computing electric flows, resistive embeddings, as well as heat kernels. In Section 4.1, we verify that the Transformer can accurately compute top-$k$ and bottom-$k$ eigenevectors of the graph Laplacian.

4. In Section 6, we demonstrate the advantage of using the linear Transformer as a replacement for Laplacian eigenvector PE on a molecular regression benchmark using the QM9 and ZINC datasets (Ruddigkeit et al., 2012; Ramakrishnan et al., 2014; Irwin et al., 2012). After replacing the Laplacian eigenvector-based PE with the linear Transformer, and training on the regression loss, we verify that the linear Transformer ***automatically learns a good PE*** for the downstream regression task that can outperform the original Laplacian eigenvector PE by a wide margin.

## 1.2 RELATED WORK

Numerous authors have proposed different ways of adapting Transformers to graphs (Dwivedi & Bresson, 2020; Ying et al., 2021; Rampášek et al., 2022; Müller et al., 2023). A particularly promising approach is to use a suitable positional encoding to incorporate structural information in the input, examples include Laplacian eigenvectors (Dwivedi & Bresson, 2020; Kreuzer et al., 2021), heat kernel (Choromanski et al., 2022), resistance distance and commute time (Ma et al., 2023; Velingker et al., 2024; Zhang et al., 2023) and shortest path distance (Ying et al., 2021). Lim et al. (2022) designed a neural network to transform eigenvectors into an encoding that has certain invariance properties. Black et al. (2024) compared the expressivity of different PE schemes. (Srinivasan & Ribeiro, 2020) studied the relationship between PE and structural graph representations.

Kim et al. (2021) explore the possibility of using pure Transformers for graph learning. They provide both nodes and edges as input tokens, with a simple encoding scheme. They also prove that such a Transformer is as expressive as a second-order invariant graph network.

A number of works have explored the use of *learned PEs* (Mialon et al., 2021; Eliasof et al., 2023; Park et al., 2022; Ma et al., 2023; Kreuzer et al., 2021; Dwivedi et al., 2021). In particular Ma et al. (2023) is based on relative random walk probabilities (RRWP), and Kreuzer et al. (2021)'s approach is based on learned eigenfunctions. In comparison, using a linear-Transformer to learn PEs has two advantages: 1. self attention can implement operations such as orthogonalization, necessary for learning multiple eigenvectors (Lemma 6). Second, self-attention enables highly-efficient algorithms based on multiplicative polynomial expansions (Lemma 4). These enable linear Transformers to learn good PEs given *only* the incidence matrix as input.

Finally, we note several relevant papers that are unrelated to graph neural networks. A recent body of work proves the ability of Transformers to implement learning algorithms, to explain the in-context learning phenomenon (Schlag et al., 2021; Von Oswald et al., 2023). Surprisingly, our construction for Lemma 1 bears several remarkable parallels to the gradient descent construction by Von Oswald et al. (2023). We conjecture that the proof of Lemma 4 may be applicable for understanding the GD++ algorithm in the same paper. In another direction, Charton (2021) show experimentally that Transformers can compute linear algebra operations such as eigenvector decomposition. Their work requires a relatively complicated matrix encoding and a large number of parameters.

## 2 PRELIMINARIES AND NOTATION

### 2.1 GRAPHS

We use $\mathcal{G} = (\mathcal{V}, \mathcal{E})$ to denote a graph with vertex set $\mathcal{V}$ and edge set $\mathcal{E}$; let $n := |\mathcal{V}|$ and $d := |\mathcal{E}|$. We often identify the vertex $v_i$ with its index $i$ for $i = 1...n$, and the edge $e_j$ with $j$ for $j = 1...d$. In general, $\mathcal{G}$ has weighted edges, defined by $r(\cdot) : \mathcal{E} \to \mathbb{R}^+$. We let $r_j := r(e_j)$. A central object of interest is the *incidence matrix* $B \in \mathbb{R}^{n \times d}$, defined as follows: to each edge $e_j = (u \sim v) \in \mathcal{E}$, assign an arbitrary orientation, e.g. $e_j = (u \to v)$. The incidence matrix $B$ is given by

$$B_{ij} = \begin{cases} -1/\sqrt{r_j}, & \text{if exists } v \in \mathcal{V} \text{ such that } e_j = (u_i \to v) \\ +1/\sqrt{r_j}, & \text{if exists } v \in \mathcal{V} \text{ such that } e_j = (v \to u_i) \\ 0, & \text{otherwise.} \end{cases} \tag{1}$$

We define the graph Laplacian as $\mathcal{L} := BB^\top \in \mathbb{R}^{n \times n}$. We use $\lambda_{\max}$ to denote the maximum eigenvalue of $\mathcal{L}$. Note that $\mathcal{L}$ always has 0 as its smallest eigenvalue, with corresponding eigenvector $\frac{1}{\sqrt{n}}\vec{1}$, the all-ones vector. This fact can be verified by noticing that each column of $B$ sums to 0. For a connected graph (as we will assume is the case throughout the paper), the second-smallest eigenvalue is always non-zero, and we will denote it as $\lambda_{\min}$. Finally, we define $\hat{I}_{n \times n} := I_{n \times n} - \frac{1}{n}\vec{1}\vec{1}^\top$. $\hat{I}_{n \times n}$ is the projection onto span($\mathcal{L}$), and functions as the identity matrix when working within span($\mathcal{L}$).

### 2.2 LINEAR TRANSFORMER

We will use $Z_0 \in \mathbb{R}^{h \times n}$ to denote the input to the Transformer. $Z_0$ encodes a graph $\mathcal{G}$, and each column of $Z_0$ encodes a single vertex in $h$ dimensions. Let $W^Q, W^K, W^V \in \mathbb{R}^{h \times h}$ denote the key, query and value parameter matrices. We define linear attention Attn as

$$\text{Attn}_{W^V, W^Q, W^K}(Z) := W^V Z Z^\top W^{Q^\top} W^K Z. \tag{2}$$

Unlike standard attention, (2) does not contain softmax activation. We construct an $L$-layer Transformer by stacking $L$ layers of the attention module (augmented with a linear operation on the skip connection). To be precise, let $Z_l$ denote the output of the $(l-1)^{th}$ layer of the Transformer. Then

$$Z_{l+1} := Z_l + \text{Attn}_{W_l^V, W_l^Q, W_l^K}(Z_l) + W^R Z_l, \tag{3}$$

where $W_l^V, W_l^Q, W_l^K$ are the value, query and key weight matrices of the linear attention module at layer $l$, and $W^R \in \mathbb{R}^{h \times h}$ is the weight matrix of the linear operation. Henceforth, we let $W^V := \{W_l^V\}_{l=0...L}, W^Q := \{W_l^Q\}_{l=0...L}, W_l^K := \{W_l^K\}_{l=0...L}, W_l^R := \{W_l^R\}_{l=0...L}$ denote collections of the parameters across all layers of an $L$-layer Transformer. Finally, we will often need to refer to specific rows of $Z_l$. We will use $[Z_l]_{i...j}$ to denote rows $i$ to $j$ of $Z_l$.

## 3 TRANSFORMERS AS POWERFUL SOLVERS FOR LAPLACIAN PROBLEMS

In this section, we discuss the capacity of the linear Transformer 3 to solve certain classes of canonical graph problems. We begin with the problem of Electric Flow in Section 3.2, where the constructed Transformer can be interpreted as implementing steps of gradient descent with respect to the energy of the induced flow. Subsequently, in Section 3.3, we provide constructions for computing the resistive embedding, as well as the heat kernel, based on implementing a truncated power series. Finally, in Section 3.4, we provide faster alternative constructions for solving electric flow and computing heat kernels, based on implementing a *multiplicative polynomial expansion*. In each case, we bound the error of the Transformer by the convergence rate of the underlying algorithms.

### 3.1 ADDITIONAL SETUP

We introduce some additional setup that is common to many lemmas in this section. We will generally consider an $L$-layer Transformer, as defined in (3), for some arbitrary $L \in \mathbb{Z}^+$. As in (3), we use $Z_l$ to denote the input to layer $l$. The input $Z_0$ will encode information about a graph $\mathcal{G}$, along with a number of demand vectors $\psi_1 \dots \psi_k \in \mathbb{R}^n$. We use $\Psi$ to denote the $n \times k$ matrix whose $i^{th}$ column is $\psi_i$. Unless otherwise stated, $Z_l \in \mathbb{R}^{(d+2k) \times n}$, where $n$ is the number of nodes, $d$ is the number of edges, and $k$ is the number of demands/queries. $Z_0^\top := [B, \quad \Psi, \quad 0_{n \times k}]$.

**On parameter size:** In a straightforward implementation, the above Transformer has feature dimension $h = (d + 2k)$. The size of $W^Q, W^K, W^V, W^R$ is $O(h^2) = O(d^2 + k^2)$, which is prohibitively large as $d$ can itself be $O(n^2)$. The size of parameter matrices can be significantly reduced to $O(k^2)$ by imposing certain constraints on the parameter matrices; we present this reduction in (7) in Section 5. For simplicity of exposition, lemmas in this section will use the naive implementation in (3). We verify later that all the constructions presented in this section can also be realized in (7).

### 3.2 SOLVING ELECTRIC FLOW WITH GRADIENT DESCENT

Assume we are given a graph $\mathcal{G} = (\mathcal{V}, \mathcal{E})$, along with a non-negative vector of resistances $r \in \mathbb{R}_+^d$. Let $R$ be the $d \times d$ diagonal matrix with $r$ on its diagonal. A flow is represented by $f \in \mathbb{R}^d$, where $f_j$ denotes the (directed) flow on edge $e_j$. The energy of an electric flow is given by $\sum_{j=1}^d r_j f_j^2$. Let $\psi \in \mathbb{R}^n$ denote a *vector of demands*. Throughout this paper, we will assume that the demand vectors satisfy *flow conservation*, i.e., $\langle \psi, \vec{1} \rangle = 0$. The $\psi$-*electric flow* is the unique minimizer of the following (primal) flow-optimization problem (by convex duality, this is equivalent to a dual potential-optimization problem):

$$\text{(primal)} \qquad \min_{f \in \mathbb{R}^d} \quad \sum_{j=1}^d r_j f_j^2 \qquad \text{subject to the constraint} \quad BR^{1/2} f = \psi. \qquad (4)$$

$$\text{(dual)} \qquad - \min_{\phi \in \mathbb{R}^n} \phi^\top \mathcal{L} \phi - 2\phi^\top \psi. \qquad (5)$$

The argument is standard; for completeness, we provide a proof of equivalence between (4) and (5) in (8) in Appendix 10.1. It follows that the optimizer $\phi^*$ of (5) has closed-form solution $\phi^* = \mathcal{L}^\dagger \psi$. In Lemma 1 below, we show a simple construction that enables the Transformer in (3) to compute in parallel, the optimal potential assignments for a set of $k$ demands $\{\psi_i\}_{i=1\dots k}$, where $\psi_i \in \mathbb{R}^n$.

**Motivation: Effective Resistance Metric**
An important practical motivation for solving the electric flow (or equivalently computing $\mathcal{L}^\dagger$) is to obtain the Effective Resistance matrix $\mathcal{R} \in \mathbb{R}^{n \times n}$. GNNs that use positional encodings derived from $\mathcal{R}$ have demonstrated state-of-the-art performance on numerous tasks, and can be shown to have good theoretical expressivity (Zhang et al., 2023; Velingker et al., 2024; Black et al., 2024).

Formally, $\mathcal{R}$ is defined as $\mathcal{R}_{ij} := (u_i - u_j)^\top \mathcal{L}^\dagger (u_i - u_j)$, where $u_i$ denotes the vector that has a 1 in the $i^{th}$ coordinate, and 0s everywhere else. Intuitively, $\mathcal{R}_{ij}$ is the potential drop required to send 1-unit of electric flow from node $i$ to node $j$. Let $\ell \in \mathbb{R}^n$ denote the vector of diagonals of $\mathcal{L}^\dagger$ (i.e. $\ell_i := \mathcal{L}_{ii}$). Then $\mathcal{R} = \vec{1}\ell^\top + \ell\vec{1}^\top - 2\mathcal{L}^\dagger$; thus computing $\mathcal{L}^\dagger$ essentially also computes $\mathcal{R}$.

**Lemma 1 (Transformer solves Electric Flow by implementing Gradient Descent)** *Consider the setup in Section 3.1. Assume that $\langle \psi_i, \vec{1} \rangle = 0$ for each $i = 1...k$. For any $\delta > 0$ and for any*

*L-layer Transformer, there exists a choice of weights $W^V, W^Q, W^K, W^R$, such that each layer of the Transformer* (3) *implements a step of gradient descent with respect to the dual electric flow objective in* (5), *with stepsize $\delta$. Consequently, the following holds for all $i = 1...k$ and for any graph Laplacian with maximum eigenvalue $\lambda_{\max} \leq 1/\delta$ and minimum nontrivial eigenvalue $\lambda_{\min}$:*

$$\left\| [Z_L]_{d+k+i}^\top - \mathcal{L}^\dagger \psi_i \right\|_2 \leq \frac{\exp\left(-\delta L \lambda_{\min}/2\right)}{\sqrt{\lambda_{\min}}} \|\psi_i\|_2.$$

**Discussion.** In the special case when $k = n$ and $\Psi := I_{n \times n} - \frac{1}{n}\vec{1}\vec{1}^\top$, $[Z_L]_{d+k+1...d+2k} \approx \mathcal{L}^\dagger$. An $\varepsilon$-approximate solution requires $L = O(\log(1/\varepsilon))$ layers; this is exactly the convergence rate of gradient descent on a Lipschitz smooth and strongly convex function. We defer details of the proof of Lemma 1 to Appendix 10.1, and we provide the explicit choice of weight matrices in (10). On a high level, we show that a *single attention layer* can implement exactly *a single gradient descent step wrt the dual objective $F_\psi$* (5), i.e. $[Z_{l+1}]_{d+k+i}^\top := [Z_l]_{d+k+i}^\top - \delta \nabla F_{\psi_i}([Z_l]_{d+k+i}^\top)$. The $\log(1/\varepsilon)$ rate then follows immediately from the convergence rate of gradient descent, combined with smoothness and (restricted) strong convex of $F_\psi$. In Lemma 4 below, we show an alternate construction that reduces this to $O(\log \log(1/\varepsilon))$.

We highlight that the constructed weight matrices are very sparse; each of $W^V, W^Q, W^K, W^R$ contains a single identity matrix in a sub-block. This sparse structure makes it possible to drastically reduce the number of parameters needed, which we exploit in Section 5 to design a more parameter efficient Transformer. We provide experimental validation of Lemma 1 in Section 3.5.

### 3.3 IMPLEMENTING TRUNCATED POWER SERIES

In this section, we present two more constructions: one for computing $\sqrt{\mathcal{L}^\dagger}$ (Lemma 2), and one for computing the heat kernel $e^{-s\mathcal{L}}$ (Lemma 3). Both quantities have been successfully used for positional encoding in various GNNs. The constructions proposed in these two lemmas are also similar, and involve implementing the power series of the respective targets.

#### 3.3.1 COMPUTING THE PRINCIPAL SQUARE ROOT $\sqrt{\mathcal{L}^\dagger}$

**Motivation : Resistive Embedding**
The following fact relates the effective resistance matrix $\mathcal{R}$ to any "square root" of $\mathcal{L}^\dagger$:

**Fact 1** *Let $\mathcal{M}$ denote any matrix that satisfies $\mathcal{M}\mathcal{M}^\top = \mathcal{L}^\dagger$. Let $\mathcal{R} \in \mathbb{R}_+^{n \times n}$ denote the matrix of effective resistances (see Section 3.2). Then $\mathcal{R}_{ij} = \|\mathcal{M}_i - \mathcal{M}_j\|_2^2$, where $\mathcal{M}_i$ is the $i^{th}$ row of $\mathcal{M}$.*

One can verify the above by noticing that $\|\mathcal{M}_i - \mathcal{M}_j\|_2^2 = \mathcal{M}_i^\top \mathcal{M}_i + \mathcal{M}_j^\top \mathcal{M}_j - 2\mathcal{M}_i^\top \mathcal{M}_j = [\mathcal{L}]_{ii} + [\mathcal{L}]_{jj} - 2\mathcal{L}_{ij}$. In some settings, it is more natural to use the rows of $\mathcal{M}$ to embed node position, instead of directly using $\mathcal{R}$: By assigning an embedding vector $w_i := \mathcal{M}_i$ to vertex $i$, the Euclidean distance between $w_i$ and $w_j$ equals the resistance distance. The matrix $\mathcal{M}$ is underdetermined, and for any $m > n$, there are infinitely many choices of $\mathcal{M}$ that satisfy $\mathcal{M}\mathcal{M}^\top = \mathcal{L}^\dagger$. Fact 1 applies to any such $\mathcal{M}$. Velingker et al. (2024) uses the rows of $\mathcal{M} = \mathcal{L}^\dagger B$ for resistive embedding. Under this choice, $\mathcal{M}_i$ has dimension $d = |\mathcal{E}|$, which can be quite large. To deal with this, they additionally performs a dimension-reduction step using Johnson Lidenstrauss.

Among all valid choices of $\mathcal{M}$, there is a unique choice that is symmetric and minimizes $\|\mathcal{M}\|_F$, namely, $US^{-1/2}U^\top$, where $USU^\top = \mathcal{L}$ is the eigenvector decomposition of $\mathcal{L}$. We reserve $\sqrt{\mathcal{L}^\dagger}$ to denote this matrix; $\sqrt{\mathcal{L}^\dagger}$ is called the *principal square root of $\mathcal{L}^\dagger$*. In practice, $\sqrt{\mathcal{L}^\dagger}$ might be preferable to, say, $\mathcal{L}^\dagger B$ because it has an embedding dimension of $n$, as opposed to the possibly much larger $d$. We present in Lemma 2 a Transformer construction for computing $\sqrt{\mathcal{L}^\dagger}$.

**Lemma 2 (Principal Square Root $\sqrt{\mathcal{L}^\dagger}$)** *Consider the setup in Section 3.1. Assume that $\psi_1...\psi_k \in \mathbb{R}^n$ satisfy $\langle \psi_i, \vec{1}\rangle = 0$. For any L-layer Transformer* (3)*, there exists a configuration of weights $W^V, W^Q, W^K, W^R$ such that the following holds: For any graph with maximum Laplacian eigenvalue less than $\lambda_{\max}$ and minimum non-trivial Laplacian eigenvalue greater than $\lambda_{\min}$, and for all $i = 1...k$:*

$$\left\| [Z_L]_{d+k+i}^\top - \sqrt{\mathcal{L}^\dagger}\psi_i \right\|_2 \leq \frac{e^{-L\lambda_{\min}/\lambda_{\max}}}{\lambda_{\min}\sqrt{L/\lambda_{\max}}} \|\psi_i\|_2.$$

**Discussion.** We defer a proof of Lemma 2 to Appendix 10.2. The high-level idea is that each layer of the Transformer implements one additional term of the power series expansion of $\sqrt{\mathcal{L}^\dagger}$. Under the choice $k = n$ and $\Psi = I_{n \times n} - (1/n)\vec{1}\vec{1}^\top$, $[Z_L]_{d+k+1...d+2k} \approx \sqrt{\mathcal{L}^\dagger}$. An $\varepsilon$ approximation requires $\log(1/\varepsilon)$ layers. We consider the more general setup involving arbitrary $\psi_i's$ as they are useful for projecting the resistive embedding onto a lower-dimensional space; this is relevant when $\psi_i$'s are trainable parameters (see e.g., the setup in Appendix 14.3). We empirically validate Lemma 2 in Section 3.5.

### 3.3.2 COMPUTING THE HEAT KERNEL: $\exp(-s\mathcal{L})$

Finally, we present a result on learning heat kernels. The heat kernel has connections to random walks and diffusion maps Coifman & Lafon (2006). It plays a central role in semi-supervised learning on graphs (Xu et al., 2020); it is also used for positional encoding (Choromanski et al., 2022). Note that sometimes, the heat kernel excludes the nullspace of $\mathcal{L}$ and is defined as $e^{-s\mathcal{L}} - \vec{1}\vec{1}^\top/n$.

**Lemma 3 (Heat Kernel $e^{-s\mathcal{L}}$)** *Consider the setup in Section 3.1. Assume that $\psi_1...\psi_k \in \mathbb{R}^n$ satisfy $\langle \psi_i, \vec{1} \rangle = 0$. Let $s > 0$ be an arbitrary temperature parameter. There exists a configuration of weights for the $L$-layer Transformer* (3) *such that the following holds: for any input graph whose Laplacian $\mathcal{L}$ satisfies $8s\lambda_{\max} \leq L$, and for all $i = 1...k$,*

$$\left\| [Z]_{L,i}^\top - e^{-s\mathcal{L}}\psi_i \right\|_2 \leq 2^{-L+8s\lambda_{\max}+1}\|\psi_i\|_2$$

**Discussion.** We defer the proof of Lemma 3 to Appendix 10.3. To obtain $\varepsilon$ approximation error, we need number of layers $L \geq O(\log(1/\varepsilon) + s\lambda_{\max})$. As with the preceding lemmas, the flexibility of choosing any number of $\psi_i$'s enables the Transformer to learn a low-dimensional projection of the heat kernel. The dependence on $s\lambda_{\max}$ is fundamental, stemming from the fact that the truncation error of the power series of $e^{-s\mathcal{L}}$ begins to shrink only after $O(s\lambda_{\max})$ the first terms. In Lemma 5 in the next section, we weaken this dependence from $s\lambda_{\max}$ to $\log(s\lambda_{\max})$. We empirically validate Lemma 3 in Section 3.5.

## 3.4 IMPLEMENTING MULTIPLICATIVE POLYNOMIAL EXPANSION

We present two alternative Transformer constructions that can achieve vastly higher accuracy than the ones in preceding sections: Lemma 4 computes an $\varepsilon$-accurate electric flow in exponentially fewer layers than Lemma 1. Lemma 5 approximates the heat kernel with higher accuracy than Lemma 3 when the number of layers is small. The key idea in both Lemmas 4 and 5 is to use the Transformer to implement a multiplicative polynomial; this in turn *makes key use of the self-attention module*.

The setup for Lemmas 4 and 5 differs in two ways from that presented in Section 3.1. First, the input to layer $l$, $Z_l$, are now in $\mathbb{R}^{3n \times n}$, instead of $\mathbb{R}^{(d+2k) \times n}$. When the graph $\mathcal{G}$ is sparse and the number of demands/queries $k$ is small, the constructions in Lemma 1 and 3 may use considerably less memory. This difference is *fundamental*, due to the storage required for raising matrix powers. Second, the input is also different; in particular, information about the graph is provided via $\mathcal{L}$ as part of the input $Z_0$, as opposed to via the incidence matrix $B^\top$ as done in Section 3.1. This difference is *not fundamental*: one can compute $\mathcal{L} = B^\top B$ from $B$ in a single attention layer; in the spirit of brevity, we omit this step. We begin with the faster construction for electric flow:

**Lemma 4** *Let $\delta > 0$. Let $Z_0^\top := \left[ (\hat{I}_{n \times n} - \delta\mathcal{L}), \quad I_{n \times n}, \quad \delta I_{n \times n} \right]$. Then there exist a choice of $W^V, W^Q, W^K, W^R$ for a $L$-layer Transformer* (3) *such that for any graph with Laplacian with smallest non-trivial eigenvalue $\lambda_{\min}$ and largest eigenvalue $\lambda_{\max} \leq 1/\delta$,*

$$\left\| [Z_L]_{2n+1...3n} - \mathcal{L}^\dagger \right\|_2 \leq \frac{1}{\lambda_{\min}} \exp\left(-\delta 2^L \lambda_{\min}\right).$$

**Discussion.** Lemma 4 shows that the Transformer can compute an $\varepsilon$-approximation to $\mathcal{L}^\dagger$ (which is sufficient but not necessary for solving arbitrary electric flow demands) using $\log\log(1/\varepsilon)$ layers. This is much fewer than the $\log(1/\varepsilon)$ layers required in Lemma 1. The key idea in the proof is to implement a multiplicative polynomial expansion for $\mathcal{L}^\dagger$. We defer the proof to Appendix 10.4.

Next, we show the alternate construction for computing the heat kernel:

**Lemma 5 (Fast Heat Kernel)** *Let $s > 0$. Let $L$ be the number of Transformer layers. Let $Z_0^\top := (I_{n \times n} - 3^{-L} s\mathcal{L})$. Then there exist a choice of $W^V, W^Q, W^K, W^R$ such that for any graph whose Laplacian $\mathcal{L}$ satisfies $s\lambda_{\max} \leq 3^L$,*

$$\|Z_L - \exp(-s\mathcal{L})\|_2 \leq 3^{-L+1} s^2 \lambda_{\max}^2.$$

**Discussion.** Lemma 5 shows that the Transformer can compute an $\varepsilon$-approximation to $e^{-s\mathcal{L}}$ using $O(\log(1/\varepsilon) + \log(s\lambda_{\max}))$ layers. The $\varepsilon$ dependence is the same as Lemma 3, but the $\log(s\lambda_{\max})$ is an improvement. When the number of layers is small, Lemma 5 gives a significantly more accurate approximation. The proof is based on the well-known approximation $e^x \approx (1 + x/L)^L$. We defer proof details for Lemma 5 to Appendix 10.4.

### 3.5 Experiments

In Figure 1, we experimentally verify that the Transformer is capable of learning to solve the three objectives presented in Lemmas 1, 2 and 3. The setup is as described in Section 3.1, with the Transformer described in (3) with $k = n$, but with the following important difference: the output of each layer is additionally normalized as follows: $[Z_l]_{1...n} \leftarrow [Z_l]_{1...n}/\|[Z_l]_{1...n}\|_F$, $[Z_l]_{n+1...n+2k} \leftarrow [Z_l]_{n+1...n+2k}/\|[Z_l]_{n+1...n+2k}\|_F$, where $\|\|_F$ is the Frobenius norm. Without this normalization, training the Transformer becomes very difficult beyond 5 layers.

We consider two kinds of random graphs: fully-connected graphs ($n = 10, d = 45$) and Circular Skip Links (CSL) graphs ($n = 10, d = 20$). Edge resistances are randomly sampled. We provide details on the sampling distributions in Appendix 13. For each input graph $\mathcal{G}$, we sample $n$ demands $\psi_1...\psi_n \in \mathbb{R}^n$ independently from the unit sphere. Let $\Psi = [\psi_1...\psi_n]$. The input to the Transformer is $\begin{bmatrix} B^\top, & \Psi^\top, & 0_{n \times n} \end{bmatrix}$, consistent with the setup described in Section 3.1. The training/test loss is given by $\text{loss}_U := \mathbb{E}\left[\frac{1}{n}\sum_{i=1}^{n} \left\|\frac{[Z_l]_{d+n+i}^\top}{\|[Z_l]_{d+n+i}\|_2} - \frac{U\psi_i}{\|U\psi_i\|_2}\right\|_2^2\right]$, where $U \in \left\{\mathcal{L}^\dagger, \sqrt{\mathcal{L}^\dagger}, e^{-0.5\mathcal{L}}\right\}$. We learn the correct solutions only up to scaling, because we need to normalize $Z_l$ per-layer. The expectation is over randomness in sampling $\mathcal{L}$ and $\Psi$. We plot the log of the respective losses against number of layers after training has converged. As can be seen, in each plot, and for both types of architectures, the loss appears to decrease exponentially with the number of layers.

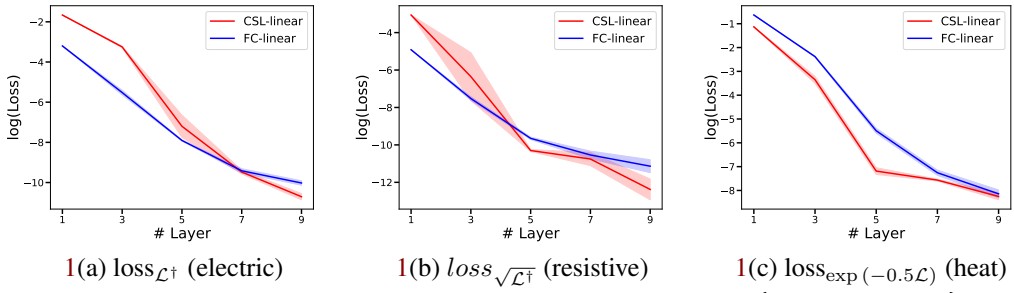

| 1(a) $\text{loss}_{\mathcal{L}^\dagger}$ (electric) | 1(b) $loss_{\sqrt{\mathcal{L}^\dagger}}$ (resistive) | 1(c) $\text{loss}_{\exp(-0.5\mathcal{L})}$ (heat) |

Figure 1: $\text{loss}_U$ against number of layers at convergence for $U \in \left\{\mathcal{L}^\dagger, \sqrt{\mathcal{L}^\dagger}, e^{-0.5\mathcal{L}}\right\}$.

## 4 Transformers can implement Subspace Iteration to compute Eigenvectors

We present a Transformer construction for computing the eigenvectors of the graph Laplacian $\mathcal{L}$. The Laplacian eigenvectors are commonly used for positional encoding (see Section 1.2). In particular, the eigenvector for the smallest non-trivial eigenvalue has been applied with great success in graph segmentation (Shi & Malik, 1997) and clustering (Bühler & Hein, 2009). Our construction is based on the *subspace iteration* algorithm (Algorithm 1), aka *block power method*—see (Bentbib & Kanber, 2015). The output $\Phi$ of Algorithm 1 converges to the top-$k$ eigenvectors of $\mathcal{L}$. In Corollary 7, we present a modified construction that instead computes the *bottom-k* eigenvectors of $\mathcal{L}$. For the purposes of this section, we consider a variant of the Transformer defined in (3).

$$\hat{Z}_{l+1} := Z_l + \text{Attn}_{W_l^V, W_l^Q, W_l^K}(Z_l) + W^R Z_l$$

$$Z_{l+1} = \text{normalize}(\hat{Z}_{l+1}), \tag{6}$$

---

**ALGORITHM 1** – Subspace Iteration

---

$\Phi_0 \in \mathbb{R}^{n \times k}$ has full column rank .
**while** not converged **do**
    $\hat{\Phi}_{i+1} \leftarrow \mathcal{L}\Phi_i$
    $\Phi_{i+1} \leftarrow QR(\hat{\Phi}_{i+1})$          ▷ QR($\Phi$) returns the Q matrix in the QR decomposition of $\Phi$.
**end while**

---

where normalize($Z$) applies row-wise normalization: $[\text{normalize}(Z)]_i \leftarrow [Z]_i / \|[Z]_i\|_2$ for $i = d + 1 \ldots d + k$. We use $B_l \in \mathbb{R}^{n \times d}$ and $\Phi_l \in \mathbb{R}^{n \times k}$ to denote refer to specific columns of $Z_l^\top$, defined as $Z_l^\top =: \begin{bmatrix} B_l^T, & \Phi_l^\top \end{bmatrix}$. The notation is chosen as $\Phi_l$ in $Z_l$ in (6) corresponds to $\Phi_l$ in Algorithm 1. We initialize $B_0 = B$ and let $\Phi_0$ be some arbitrary matrix with full column rank. $\Phi_1$ becomes orthogonal after the first QR step of Algorithm 1.

**Lemma 6 (Subspace Iteration for Finding Top $k$ Eigenvectors)** *Consider the Transformer defined in (6). There exists a choice of $W^V, W^Q, W^K, W^R$ such that $k + 1$ layers of the Transformer implements one iteration of Algorithm 1. Consequently, the output $\Phi_L$ of a $L$-layer Transformer approximates the top-$k$ eigenvectors of $\mathcal{L}$ to the same accuracy as $L/(k+1)$ steps of Algorithm 1.*

**Discussion.** We bound the Transformer's error by the error of Algorithm 1, but do not provide an explicit convergence rate. This omission is because the convergence rate of Algorithm 1 itself is difficult to characterize, and has a complicated dependence on pairwise spectral gaps of $\Phi_0$. The high-level proof idea is to use self-attention to orthogonalize the columns of $\Phi_l$ (Lemma 8. We defer the proof of Lemma 6 to Appendix 11. We experimentally validate Lemma 6 in Section 4.1.

The construction in Lemma 6 explicitly requires $k$ layers to perform a QR decomposition of $\Phi$ plus 1 layer to multiply by $\mathcal{L}$. The layer usage can be much more efficient in practice: First, multiple $\mathcal{L}$-multiplications can take place before a single QR-factorization step. Second, the $k$ layers for performing QR decomposition can be implemented in a single layer with $k$ parallel heads.

In graph applications, one is often interested in the *bottom-$k$* eigenvectors of $\mathcal{L}$. The following corollary shows that a minor modification of Lemma 6 will instead compute the bottom $k$ eigenvectors.

**Corollary 7 (Subspace Iteration for Finding Bottom $k$ Eigenvectors)** *Consider the same setup as Lemma 6. Let $\mu > 0$ be some constant. There exists a construction for a $L$-layer Transformer (similar to Lemma 6), which implements Algorithm 1 with $\mathcal{L}$ replaced by $\mu I - \mathcal{L}$, and $\Phi_L$ approximates the bottom $k$ eigenvectors of $\mathcal{L}$ if $\lambda_{\max}(\mathcal{L}) \leq \mu$.*

We provide a short proof in Appendix 11. The key idea is that a minor modification of the construction in Lemma 6 will instead compute the bottom $k$ eigenvectors. Alternative to the construction in Corollary 7, one can also first compute $\mathcal{L}^\dagger$ (via Lemma 4), followed by subspace iteration for $\mathcal{L}^\dagger$.

## 4.1 EXPERIMENTS FOR LEMMA 6

We verify Lemma 6 and Corollary 7 experimentally by evaluating the ability of the Transformer (3) to learn top-$k$ and bottom-$k$ eigenvectors. As in Section 3.5, we consider two kinds of random graphs with $n = 10$ nodes: fully connected ($d = 45$ edges) and CSL ($d = 20$ edges); each edge is has a randomly sampled resistance; see Appendix 13 for details. For a graph $\mathcal{G}$ with Laplacian $\mathcal{L}$, let $\lambda_1 \leq \lambda_2 \leq \ldots \lambda_{10}$ denote its eigenvalues. Let $v_1, \ldots, v_{10}$ denote its eigenvectors. $\lambda_1$ is always 0 and $v_1$ is always $\vec{1}/\sqrt{n}$.

The Transformer architecture is as defined in Section 6, with $k = n$. We increase the dimension of $\Phi_l$ to $(2n) \times n$, and thus the dimension of $Z_l$ to $(d + n + n) \times n$. We read out the last $n$ rows of $Z_L$ as output. The purpose of increasing the dimension is to make the architecture identical to the one used in the experiments in Section 3.5; the construction in Lemmas 6 and Corollary 7 extend to this setting by setting appropriate parameters to 0. In addition, we also normalize $[Z_l]_{1\ldots d}$ each layer by its Frobenius norm, i.e., $[Z_l]_{1\ldots d} \leftarrow [Z_l]_{1\ldots d} / \|[Z_l]_{1\ldots d}\|_F$ (the proof of Lemma 6 still holds as subspace iteration is scaling-invariant). The input to the Transformer is $Z_0^\top = [B, \quad \Phi_0]$, and we make $\Phi_0 \in \mathbb{R}^{2n \times n}$ a trainable parameter along with $W^V, W^Q, W^K, W^R$. We define $\text{loss}_i := \mathbb{E}\left[\min\left\{\|\phi_{L,i} - v_i\|_2^2, \|\phi_{L,i} + v_i\|_2^2\right\}\right]$, where $\phi_{L,i}$ is the $i^{th}$ row of $\Phi_L$ and expectation is taken with

respect to randomness in sampling the graph; the min is required because eigenvectors are invariant to sign-flips. We train and evaluate the Transformer on two kinds of losses: $\text{loss}_{1-5} := \frac{1}{5} \sum_{i=1}^{5} \text{loss}_i$ and $\text{loss}_{1-10} := \frac{1}{10} \sum_{i=1}^{10} \text{loss}_i$. We plot the results in Figure 2, and summarize our findings below:

1. 2(a) and 2(d): Both $\text{loss}_{1-5}$ and $\text{loss}_{1-10}$ appear to decrease exponentially with the number of Transformer layers, for both FC and CSL graphs. This is consistent with Lemma 6 and Corollary 7.

2. In Figures 2(e) and 2(f), when the Transformer is trained on $\text{loss}_{1-10}$, **larger eigenvectors are learned more accurately**, with the exception of $loss_2$. This is mostly consistent with our construction in Lemma 6, where the larger eigenvectors are computed more quickly.

   In contrast, in Figures 2(b) and 2(c), when the Transformer is trained on $\text{loss}_{1-5}$, the **smaller eigenvectors are learned more accurately**. This is consistent with our construction in Corollary 7, where the smaller eigenvectors of $\mathcal{L}$ are also the larger eigenvectors of $\alpha I - \mathcal{L}$, and are thus computed more quickly. One expects to observe the same ordering if the Transformer is instead computing $\mathcal{L}^\dagger$ first, followed by subspace iteration.

We omit $\text{loss}_1$ from Figures 2(b), 2(c), 2(e), 2(f) because $v_1$ is a constant so $\text{loss}_1$ goes to $0$ extremely fast, making it hard to see the other lines.

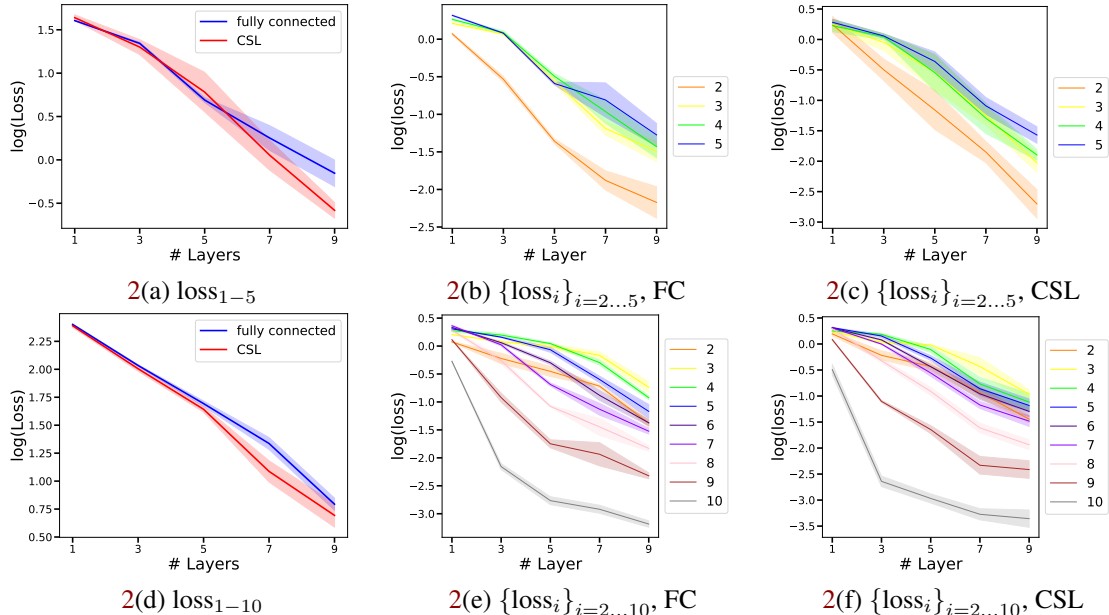

$$\text{2(a) } \text{loss}_{1-5} \qquad \text{2(b) } \{\text{loss}_i\}_{i=2\ldots5}, \text{FC} \qquad \text{2(c) } \{\text{loss}_i\}_{i=2\ldots5}, \text{CSL}$$

$$\text{2(d) } \text{loss}_{1-10} \qquad \text{2(e) } \{\text{loss}_i\}_{i=2\ldots10}, \text{FC} \qquad \text{2(f) } \{\text{loss}_i\}_{i=2\ldots10}, \text{CSL}$$

Figure 2: $\log(\text{loss}_*)$ vs. number of layers. Top row: various losses for the Transformer **trained on $\text{loss}_{1-5}$**. Bottom row: various losses for the Transformer **trained on $\text{loss}_{1-10}$**.

## 5 PARAMETER-EFFICIENT IMPLEMENTATION

As explained in Section 3.1, the sizes of the $W^Q, W^K, W^V, W^R$ matrices can scale with $O(n^4)$ in the worst case, where $n$ is the number of nodes. This makes the memory requirement prohibitive. To alleviate this, we present below in (7) a more efficient Transformer implementation than the one defined in (3). The representation in (7) is strictly more constrained than (3), but it is still expressive enough to implement all the previous constructions. For each layer $l$, let $\alpha_l^V, \alpha_l^Q, \alpha_l^K, \alpha_l^R$ be scalar weight parameters, and let $W_l^{V,\Phi}, W_l^{Q,\Phi}, W_l^{K,\Phi}, W_l^{R,\Phi} \in \mathbb{R}^{2k \times 2k}$. Let $B_l, \Phi_l$ evolve as

$$B_{l+1}^\top = (1 + \alpha_l^R)B_l^\top + \alpha_l^V B_l^\top \left( \alpha_l^Q \alpha_l^K B_l B_l^\top + \Phi_l W_l^{Q,\Phi^\top} W_l^{K,\Phi} \Phi_l^\top \right) \tag{7}$$

$$\Phi_{l+1}^\top = \left( I + W_l^{R,\Phi} \right) \Phi_l^\top + W_l^{V,\Phi} \Phi_l^\top \left( \alpha_l^Q \alpha_l^K B_l B_l^\top + \Phi_l W_l^{Q,\Phi^\top} W_l^{K,\Phi} \Phi_l^\top \right),$$

with initialization $B_0 := B$, and $\Phi_0 = [\Psi, 0_{k \times n}]$. We verify that (7) matches (3), with $Z_l = \left[ B_l^\top, \Phi_l^\top \right]$; the difference is that certain blocks of $W_l^V, W_l^Q, W_l^K, W_l^R$ constrained to be zero or

some scaling of identity. Nonetheless, we verify that the constructions in Lemmas 1, 2, 3 and 6 can all be realized within the constrained dynamics (7). We prove this in Lemma 9 in Appendix 12. In Figure 15, we show that the efficient implementation (7) performs similarly (and in many cases better than) the standard implementation (3) on all the synthetic experiments.

Besides the reduced parameter size, we highlight two additional advantages of (7):

1. Let $\mathsf{TF}_l^B(B, \Phi)$ (resp $\mathsf{TF}_l^\Phi(B, \Phi)$) be defined as the value of $B_l^\top$ (resp $\Phi_l^\top$) when initialized at $B_0 = B$ and $\Phi_0 = \Phi$ under the dynamics (7). Let $U \in \mathbb{R}^{d \times d}$ be some permutation matrix over edge indices. Then

   (a) $\mathsf{TF}_l^B$ is *equivariant* to edge permutation, i.e. $\mathsf{TF}_l^B(BU, \Phi) = U^\top \mathsf{TF}_l^B(B, \Phi)$

   (b) $\mathsf{TF}_l^\Phi$ is *invariant* to edge permutation, i.e. $\mathsf{TF}_l^\Phi(BU, \Phi) = \mathsf{TF}_l^\Phi(B, \Phi)$.

   We provide a short proof of this in Lemma 10 in Appendix 12.

2. If $B_l$ is sparse, then $B_l^\top B_l$ can be efficiently computed using sparse matrix multiply. This is the case for all of our constructions for all layers $l$.

## 6 LEARNING POSITIONAL ENCODING FOR MOLECULAR REGRESSION

In Sections 3.5 and 4.1, we saw that the Transformer is capable of learning a variety of embeddings based on $\mathcal{L}^\dagger, \sqrt{\mathcal{L}}, e^{-s\mathcal{L}}$, and $\mathrm{EVD}(\mathcal{L})$, **when the training loss is explicitly the recovery loss for that embedding**. An natural question is then: *Can a GNN perform as well on a downstream task if we replace its PE by a linear Transformer, and train the GNN together with the linear Transformer?* There are two potential benefits to this approach: First, the Transformer can **learn a PE that is better-suited to the task than hard-coded PEs, and thus achieve higher accuracy**. Second, the Transformer with a few layers/dimensions may learn to compute only the most relevant features, and achieve **comparable accuracy to the hard-coded PE using less computation.**

To test this, we evaluate the performance of our proposed linear Transformer on a molecular regression task on two real-world datasets: QM9 (Ruddigkeit et al., 2012; Ramakrishnan et al., 2014) and ZINC (Irwin et al., 2012). The regression target is constrained solubility (Dwivedi et al., 2023). Our experiments are based on the Graph Transformer (GT) implementation from Dwivedi & Bresson (2020). In Table 2, we compare three loss values: GT without PE, GT with Laplacian Eigenvector as PE (LapPE), and GT with Linear Transformer output as PE. We use LapPE as a baseline because it was the PE used in Dwivedi & Bresson (2020). We present details of the datasets in Appendix 14.1. The linear Transformer architecture is a modified version of (7), detailed in Appendix 14.2. Other experiment details, including precise model definitions, can be found in Appendix 14.3.

| | Model | # Parameters | Loss |
|---|---|---|---|
| ZINC | Graph Transformer | $800771 - 891$ | $0.286 \pm 0.0078$ |
| | Graph Transformer + LapPE | $800771$ | $0.201 \pm 0.0034$ |
| | Graph Transformer + Linear Transformer | $800771 + 488$ | $0.138 \pm 0.012$ |
| QM9 | Graph Transformer | $799747 - 512$ | $0.419 \pm 0.0047$ |
| | Graph Transformer + LapPE | $799747$ | $0.227 \pm 0.0094$ |
| | Graph Transformer + Linear Transformer | $799747 + 240$ | $0.221 \pm 0.0060$ |

Table 2: Regression Loss for ZINC and QM9 for different choices of PE.

The difference between **GT** and **GT + LapPE** is substantial for both QM9 and ZINC, highlighting the importance of PE in both cases. Going from **GT + LapPE** to **GT + Linear Transformer** for ZINC, there is a significant further improvement in loss (about $30\%$). This is remarkable, considering that the linear Transformer accounts for less than $0.7\%$ of the total number of parameters. Note that the SOTA error for ZINC regression is significantly lower than 0.138 on more recent architectures; the significance of our result is in demonstrating the improvement *just by replacing the LapPE with the linear Transformer, while keeping everything else fixed.* In contrast, going from **GT + LapPE** to **GT + Linear Transformer** for QM9, the difference is essentially zero. We conjecture that this may be because QM9 molecules (average of about 9 nodes and 19 edges) are considerably smaller than ZINC (average of about 23 nodes and 40 edges). Thus there may not be too many additional useful features to learn, and LapPE close to optimal. It is still consistent with our theory, for the Linear Transformer to do as well as LapPE.

## 7 ETHICS STATEMENT

The authors have adhered to ICLR's code of ethics during the writing of this paper, and when conducting the research described herein. There are no ethics concerns which need to be highlighted.

## 8 REPRODUCIBILITY

We use both synthetic data and open-source molecular datasets in our experiments. Details of our experiment procedures and implementations have been provided in Section 3.5, Section 4.1, Section 6 and Appendix 14.

## 9 ACKNOWLEDGEMENTS

Suvrit Sra Acknowledges generous support from the Alexander von Humboldt Foundation's AI Professorship. The authors thank Aaron Schild for the many valuable discussions, and Professor Xavier Bresson from the National University of Singapore for sharing the code which facilitated some of the experiments in this work.

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

## 10   PROOFS FOR SECTION 3

### 10.1   ELECTRIC FLOW

**Proof of Primal Dual Equivalence for Electric Flow**
By convex duality, the primal problem in (4) can be reformulated as

$$
\min_{f:BR^{1/2}f=\psi} \quad \sum_{j=1}^{d} r_j f_j^2
$$

$$
= \min_f \max_\phi f^\top R f - 2\phi^\top (BR^{1/2}f - \psi)
$$

$$
= \max_\phi \min_f f^\top R f - 2\phi^\top (BR^{1/2}f - \psi)
$$

$$
= \max_\phi -\phi^\top BB^\top \phi + 2\phi^\top \psi
$$

$$
= -\min_{\phi\in\mathbb{R}^n} \phi^\top \mathcal{L}\phi - 2\phi^\top \psi \tag{8}
$$

**Proof of Lemma 1**
Let $F_\psi(\phi) := \frac{1}{2}\phi^\top \mathcal{L}\phi - \phi^\top \psi$ denote (1/2 times) the dual optimization objective from (5). Its gradient is given by

$$
\nabla F_\psi(\phi) = \mathcal{L}\phi - \psi.
$$

Let $Z_l$ denote the output after $l$ layers of the Transformer (3), i.e. $Z_l$ evolves according to (3):

$$
Z_{l+1} := Z_l + W_l^V Z_l Z_l^\top W_l^{Q\top} W_l^K Z_l + W_l^R Z_l.
$$

We use $B_l^\top$ to denote the first $d$ rows of $Z_l$, $\Lambda_l^\top$ to denote the $(d+1)^{th}$ row to $(d+k)^{th}$ row of $Z_l$ and $\Phi_l^\top$ to denote the last $k$ rows of $Z_l$, i.e. $Z_l = \begin{bmatrix} B_l^\top \\ \Lambda_l^\top \\ \Phi_l^\top \end{bmatrix}$. For $i = 1...k$, let $\phi_{l,i}$ denote the $i^{th}$ column of $\Phi_l$. Then there exists a choice of $W^V, W^Q, W^K, W^R$, such that for all $i$,

$$
\phi_{l+1,i} = \phi_{l,i} - \delta\nabla F_{\psi_i}(\phi_{l,i}). \tag{9}
$$

Before proving (9), we first consider its consequences. We can verify that $\nabla^2 F_\psi(\cdot) = \mathcal{L}$, which is in turn upper and lower bounded by $\lambda_{\min} \prec \mathcal{L} \prec \lambda_{\max}$. By standard analysis of gradient descent for strongly convex and Lipschitz smooth functions, for $\delta < \frac{1}{\lambda_{\max}}$, we verify that for all $i = 1...k$,

$$
\|\phi_{L,i} - \mathcal{L}^\dagger \psi_i\|_2^2
$$

$$
= \|\phi_{L,i} - \phi_i^*\|_2^2
$$

$$
\leq (1 - \delta\lambda_{\min})\|\phi_{L-1,i} - \phi_i^*\|_2^2
$$

$$
\leq e^{-\delta L\lambda_{\min}}\|\phi_{0,i} - \phi_i^*\|_2^2
$$

$$
= e^{-\delta L\lambda_{\min}}\|\phi_i^*\|_2^2
$$

$$
\leq \frac{\exp(-\delta L\lambda_{\min})}{\lambda_{\min}}\|\psi\|_2^2
$$

where $\phi_i^* := \arg\min_\phi F_{\psi_i}(\phi) = \mathcal{L}^\dagger \psi_i$.

As a final note, $F$ is only weakly convex along the $\vec{1}$ direction, but we can ignore this because $\phi_{l,i}$ will always be orthogonal to $\vec{1}$ as long as $\phi_{0,i}$ is (as we assumed in the lemma statement).

The remainder of the proof will be devoted to showing (9). Recall that the input to the Transformer $Z_0$ is initialized as $Z_0 = \begin{bmatrix} B^\top \\ \Psi^\top \\ 0_{k\times n} \end{bmatrix} \in \mathbb{R}^{(d+2k)\times n}$. For some fixed $\delta$, our choice of parameter matrices

will be the same across layers:

$$W_l^V = \begin{bmatrix} 0_{d\times d} & 0_{d\times k} & 0_{d\times k} \\ 0_{k\times d} & 0_{k\times k} & 0_{k\times k} \\ 0_{k\times d} & 0_{k\times k} & -\delta I_{k\times k} \end{bmatrix}, \; W_l^{Q\top} W_l^K = \begin{bmatrix} I_{d\times d} & 0_{d\times k} & 0_{d\times k} \\ 0_{k\times d} & 0_{k\times k} & 0_{k\times k} \\ 0_{k\times d} & 0_{k\times k} & 0_{k\times k} \end{bmatrix}, \; W_l^R = \begin{bmatrix} 0_{d\times d} & 0_{d\times k} & 0_{d\times k} \\ 0_{k\times d} & 0_{k\times k} & 0_{k\times k} \\ 0_{k\times d} & \delta I_{k\times k} & 0_{k\times k} \end{bmatrix}$$

$$(10)$$

By the choice of $W^V, W^R$ in (10), we verify that for all layers $l$, $B_l = B_0 = B$, so that $Z_l^\top (W_l^Q)^\top W_l^K Z_l = \mathcal{L}$. Again by choice of $W^V, W^R$ in (10), $\Lambda_l = \Psi$ for all $l$. It thus follows from induction that

$$\Phi_{l+1} = \Phi_l - \delta\mathcal{L}\Phi_l + \delta\Psi \tag{11}$$
$$\Leftrightarrow \quad \phi_{l+1,i} = \phi_{l,i} - \delta\mathcal{L}\phi_{l,i} + \delta\psi_i = \phi_{l,i} - \delta\nabla F_{\psi_i}(\phi_{l,i}) \qquad \text{for } i = 1...k,$$

thus proving (9).

## 10.2 RESISTIVE EMBEDDING

**Proof of Lemma 2**

Let $\hat{I} := I_{n\times n} - \frac{1}{n}\vec{1}\vec{1}^\top$. Let $\delta := 1/\lambda_{\max}$.

Consider the matrix power series, which converges under our choice of $\delta$:

$$\sqrt{\mathcal{L}^\dagger} = \sqrt{\delta}\sqrt{\left(\hat{I} - (\hat{I} - \delta\mathcal{L})\right)^\dagger} = \sqrt{\delta}\sum_{l=0}^{\infty}\binom{2l}{l}\frac{(I - \delta\mathcal{L})^l}{4^l}\hat{I},$$

where the second inequality uses the fact that $(I - \delta\mathcal{L})\hat{I} = \left(\hat{I} - \delta\mathcal{L}\right)\hat{I}$.

We define $\alpha_l := \sqrt{\delta}\binom{2l}{l}\frac{1}{4^l}$.

$$W_l^V = \begin{bmatrix} 0_{d\times d} & 0_{d\times k} & 0_{d\times k} \\ 0_{k\times d} & -\delta I_{k\times k} & 0_{k\times k} \\ 0_{k\times d} & 0_{k\times k} & 0_{k\times k} \end{bmatrix}, \; W_l^{Q\top} W_l^K = \begin{bmatrix} I_{d\times d} & 0_{d\times k} & 0_{d\times k} \\ 0_{k\times d} & 0_{k\times k} & 0_{k\times k} \\ 0_{k\times d} & 0_{k\times k} & 0_{k\times k} \end{bmatrix}, \; W_l^R = \begin{bmatrix} 0_{d\times d} & 0_{d\times k} & 0_{d\times k} \\ 0_{k\times d} & 0_{k\times k} & 0_{k\times k} \\ 0_{k\times d} & \alpha_l I_{k\times k} & 0_{k\times k} \end{bmatrix}$$

$$(12)$$

Define $Z_l =: \begin{bmatrix} B_l^\top \\ \Lambda_l \\ \Phi_l^\top \end{bmatrix}$. We verify that for all $l$, $B_l = B$, and

$$\Lambda_{l+1} = (I - \delta\mathcal{L})\Lambda_l = (I - \delta\mathcal{L})^l$$

$$\Phi_{l+1} = \Phi_l + \alpha_l\Lambda_l = \sum_{i=0}^{l}\alpha_i(I - \delta\mathcal{L})^i\hat{I}\Psi,$$

where we use the fact that $\Psi = \hat{I}\Psi$ by assumption. Therefore,

$$\left\|[Z]_{L,i}^\top - \sqrt{\mathcal{L}^\dagger}\psi_i\right\|_2 \leq \left\|\sum_{i=L}^{\infty}\alpha_i(\hat{I} - \delta\mathcal{L})^i\right\|_2 \|\psi_i\|_2$$

By upper and lower bounds of Stirling's formula $\sqrt{2\pi L}L^L e^{-L} \leq L! \leq e\sqrt{L}L^L e^{-L}$, we can bound $\alpha_L \leq \sqrt{\delta/L}$. Using the fact that $(\hat{I} - \delta\mathcal{L}) \prec (I - \delta\lambda_{\min})\hat{I} \prec e^{-\delta\lambda_{\min}}\hat{I}$, and using the bound on geometric sum, the above is bounded by $\frac{e^{-L\delta\lambda_{\min}}}{\lambda_{\min}\sqrt{\delta L}}\|\psi_i\|_2$. The conclusion follows by plugging in $\delta = 1/\lambda_{\max}$.

## 10.3 HEAT KERNEL

**Proof of Lemma 3**

The power series for $e^{-s\mathcal{L}}$ is given by

$$e^{-s\mathcal{L}} = \sum_{l=0}^{\infty}\frac{(-s)^l\mathcal{L}^l}{l!}.$$

We define $\alpha_l := \frac{(-s)^l}{l!}$. For each layer $l$, define

$$W_l^V = \begin{bmatrix} 0_{d\times d} & 0_{d\times k} & 0_{d\times k} \\ 0_{k\times d} & I_{k\times k} & 0_{k\times k} \\ 0_{k\times d} & 0_{k\times k} & 0_{k\times k} \end{bmatrix}, \ W_l^{Q\top} W_l^K = \begin{bmatrix} I_{d\times d} & 0_{d\times k} & 0_{d\times k} \\ 0_{k\times d} & 0_{k\times k} & 0_{k\times k} \\ 0_{k\times d} & 0_{k\times k} & 0_{k\times k} \end{bmatrix}, \ W_l^R = \begin{bmatrix} 0_{d\times d} & 0_{d\times k} & 0_{d\times k} \\ 0_{k\times d} & -I_{k\times k} & 0_{k\times k} \\ 0_{k\times d} & \alpha_l I_{k\times k} & 0_{k\times k} \end{bmatrix}$$

Define $Z_l =: \begin{bmatrix} B_l^\top \\ \Lambda_l \\ \Phi_l^\top \end{bmatrix}$. We verify that for all $l$, $B_l = B$, and

$$\Lambda_{l+1} = \mathcal{L}\Lambda_l = \mathcal{L}^{l+1}\Psi$$

$$\Phi_{l+1} = \Phi_l + \alpha_l \Lambda_l = \sum_{i=0}^{l} \alpha_i \mathcal{L}^i \Psi.$$

Therefore,

$$\left\| [Z]_{L,i}^\top - e^{-s\mathcal{L}}\psi_i \right\|_2 \leq \left\| \sum_{i=L+1}^{\infty} \alpha_i \mathcal{L}^i \right\|_2 \|\psi_i\|_2$$

By lower bound of Stirling's formula, $L! \geq (L/e)^L$. Therefore, for $L \geq 8s\lambda_{\max}$, we have $\left\| \alpha_L \mathcal{L}^L \right\|_2 \leq 2^{-L+8s\lambda_{\max}}$. For $L \geq 2$, the infinite sum is within a factor 2 of the first term, thus $\left\| \sum_{i=L+1}^{\infty} \alpha_i \mathcal{L}^i \right\|_2 \leq 2^{-L+8s\lambda_{\max}+1}$.

## 10.4 Faster Constructions

**Proof of Lemma 4**
We begin by recalling the Taylor expansion of $\mathcal{L}^\dagger$:

$$\mathcal{L}^\dagger = \delta \sum_{l=0}^{\infty} \left( \hat{I} - \delta\mathcal{L} \right)^l.$$

(recall that $\hat{I} := I_{n\times n} - \vec{1}\vec{1}^\top / n$). Our proof of this section uses a simple but powerful alternative expansion (see for example Peng & Spielman (2014)): for any integer $t$,

$$\delta \prod_{l=0}^{t} \left( I + \left( \hat{I} - \delta\mathcal{L} \right)^{2^l} \right) = \delta \sum_{l=0}^{2^t} (\hat{I} - \delta\mathcal{L})^l. \tag{13}$$

Notice that $t$ terms in the LHS equals $2^t$ terms of the RHS. This is exactly why we will get a doubly exponential convergence. As seen in the proof of Lemma 1, each term in the RHS of (13) coincides with one step of gradient descent. Therefore, if one layer of the Transformer can implement one additional product on the LHS, a $L$ layer Transformer can efficiently implement $2^L$ steps of gradient descent. In the remainder of the proof, we will show exactly this. For all layers $l$, let

$$W_l^V = \begin{bmatrix} I_{n\times n} & 0_{n\times n} & 0_{n\times n} \\ 0_{n\times n} & 0_{n\times n} & 0_{n\times n} \\ I_{n\times n} & 0_{n\times n} & 0_{n\times n} \end{bmatrix}, \ W_l^{Q\top} W_l^K = \begin{bmatrix} 0_{n\times n} & 0_{n\times n} & 0_{n\times n} \\ I_{n\times n} & 0_{n\times n} & 0_{n\times n} \\ 0_{n\times n} & 0_{n\times n} & 0_{n\times n} \end{bmatrix}, \ W_l^R = \begin{bmatrix} -I_{n\times n} & 0_{n\times n} & 0_{n\times n} \\ 0_{n\times n} & 0_{n\times n} & 0_{n\times n} \\ 0_{n\times n} & 0_{n\times n} & 0_{n\times n} \end{bmatrix}.$$

Let $Z_l^\top =: [\Gamma_l, \quad \Lambda_l, \quad \Phi_l]$, where $\Gamma_l, \Lambda_l, \Phi_l \in \mathbb{R}^{n\times n}$. Under the configuration of weight matrices above, we verify that $\Lambda_l = I_{n\times n}$ for all $l$, and thus $Z_l^\top W_l^{Q\top} W_l^K Z_l = \Lambda_l \Gamma_l^\top = \Gamma_l$. for all $l$.

Next, we verify by induction that $\Gamma_{l+1} = \Gamma_l - \Gamma_l + \Gamma_l^\top \Gamma_l = \Gamma_l^\top \Gamma_l = \left( \hat{I} - \delta\mathcal{L} \right)^{2^l}$. Note that $\Gamma_l$ is symmetric.

Finally, we verify that $\Phi_{l+1} = \Phi_l + \Gamma_l^\top \Phi_l = \left( I + \left( \hat{I} - \delta\mathcal{L} \right)^{2^l} \right) \Phi_l$. Thus by induction,

$$\Phi_L = \delta \prod_{i=0}^{L} \left( I + (\hat{I} - \delta\mathcal{L})^{2^i} \right).$$

Finally, again using (13), the residual term is given by

$$\mathcal{L}^\dagger - \Phi_L = \delta \sum_{i=2^L+1}^{\infty} \left(\hat{I} - \delta\mathcal{L}\right)^i.$$

Noting that $(\hat{I} - \delta\mathcal{L}) \prec (1 - \delta\lambda_{\min}\mathcal{L})\hat{I}$, we can bound

$$\left\|\mathcal{L}^\dagger - \Phi_L\right\|_2 \leq \frac{\exp\left(-2^L\delta\lambda_{\min}\right)}{\lambda_{\min}}.$$

**Proof of Lemma 5**

Let $C := 3^L$. We will use the bound

$$(I - s\mathcal{L}/C)^C \prec e^{-s\mathcal{L}} \prec (I - s\mathcal{L}/C)^C\left(I - s^2\mathcal{L}^2/C\right)^{-1} \prec (I - s\mathcal{L}/C)^C\left(I + 2s^2\mathcal{L}^2/C\right),$$

where the second inequality holds by our assumption that $s\lambda_{\max} < 1/C$. Therefore,

$$\left\|(I - s\mathcal{L}/C)^C - \exp(-s\mathcal{L})\right\|_2 \leq \frac{2s^2\mathcal{L}^2}{C}\left\|\exp(-s\mathcal{L})\right\|_2 \leq \frac{2s^2\lambda_{\max}^2}{C}.$$

We will now show that $Z_L = (I - s\mathcal{L}/C)^C$. Let us define, for all $l$,

$$W_l^V = I_{n\times n}, \qquad W_l^{Q^\top}W_l^K = I_{n\times n}, \qquad W_l^R = -I_{n\times n}.$$

Then we verify that $Z_{l+1} = Z_l - Z_l + Z_l Z_l^\top Z_l = Z_l^3$ (by symmetry of $Z_l$). Thus by induction, $Z_l = Z_0^{3^l} = (I - s\mathcal{L}/C)^{3^l} = (I - s\mathcal{L}/C)^C$.

## 11 PROOFS FOR SECTION 4

**Lemma 8 (Single Index Orthogonalization)** *Consider the same setup as Lemma 6, with Transformer defined in (6). Let $\phi_{l,j} \in \mathbb{R}^n$ denote the $j^{th}$ column of $\Phi_l$. For any $i = 1...k$, there exists a choice of $W^V, W^Q, W^K, W^R$, such that*

$$\hat{\phi}_{l+1,i} = \phi_{l,i} - \sum_{j=i+1}^{k} \langle \phi_{l,i}, \phi_{l,j}\rangle \phi_{l,j}, \qquad \phi_{l+1,i} = \frac{\hat{\phi}_{l+1,i}}{\left\|\hat{\phi}_{l+1,i}\right\|_2},$$

*and for any $j \neq i$,*

$$\phi_{l+1,j} = \phi_{l,j}.$$

**Proof of Lemma 8**

Recall that $Z_l^\top =: [B_l, \quad \Phi_l]$. Let $A \in \mathbb{R}^{k\times k}$ denote the matrix where $A_{ii} = 1$ and is $0$ everywhere else. Let $H$ denote the matrix where $H_{jj} = 1$ for all $j > i$, and is $0$ everywhere else. Let the weight matrices be defined as

$$W_l^V = -\begin{bmatrix} 0_{d\times d} & 0_{d\times k} \\ 0_{k\times d} & A \end{bmatrix}, \qquad W_l^{Q^\top}W_l^K = \begin{bmatrix} 0_{d\times d} & 0_{d\times k} \\ 0_{k\times d} & H \end{bmatrix}, \qquad W^R = 0$$

Under the above definition, $B_l^\top = B^\top$ is a constant across all layers $l$. We verify that $\Phi_l$ evolves as

$$\Phi_{l+1}^\top = \Phi_l^\top - A\Phi_l^\top \Phi_l H\Phi_l^\top.$$

We verify that the effect of left-multiplication by $A$ "selects the $i^{th}$ row of $\Phi_l^\top\Phi_l$" and the effect of right-multiplication by $H$ "selects the $(i+1)^{th}...k^{th}$ columns of $\Phi_l^\top\Phi_l$". Thus

$$A\Phi^\top\Phi H = \begin{bmatrix} 0 & ... & 0 & ... & 0 \\ \vdots & & \vdots & & \vdots \\ 0 & ... & \langle\phi_{l,i},\phi_{l,j+1}\rangle & ... & \langle\phi_{l,i},\phi_{l,k}\rangle \\ \vdots & & \vdots & & \vdots \\ 0 & ... & 0 & ... & 0 \end{bmatrix}.$$

The conclusion can be verified by right-multiplying the above matrix with $\Phi_l^\top$, and then applying columnn-wise normalization to $\Phi_l$ (or equivalently, applying row-wise normalization to $[Z]_{d...d+k}$), as described in (6).

**Proof of Lemma 6**

Recall that $Z_l =: \begin{bmatrix} B_l^T \\ \Phi_l^\top \end{bmatrix}$. Assume for now that $B_l = B$ for all layers $l$. Let $\phi_{l,i}$ denote the $i^{th}$ column of $\Phi_l$. Let $l$ be some fixed layer with weights

$$W_l^V = \begin{bmatrix} 0_{d\times d} & 0_{d\times k} \\ 0_{k\times d} & I_{k\times k} \end{bmatrix}, \qquad W_l^{Q\top} W_l^K = \begin{bmatrix} I_{d\times d} & 0_{d\times k} \\ 0_{k\times d} & 0_{k\times k} \end{bmatrix}, \qquad W_l^R = \begin{bmatrix} I_{d\times d} & 0_{d\times k} \\ 0_{k\times d} & -I_{k\times k} \end{bmatrix}. \quad (14)$$

Combined with (6), these imply that $\hat{\Phi}_{l+1} = \mathcal{L}\Phi_l$, and that $\phi_{l+1,i} = \phi_{l+1,i}/\|\phi_{l+1,i}\|_2$. This implements the first step in the loop of Algorithm 1. Although Algorithm 1 does not contain the normalization step, the result is identical, because $QR(\Phi)$ is invariant to column-scaling of $\Phi$.

We now show a (different) configuration of weight matrices which enable a sequence of layers to perform QR decomposition: Consider the construction in Lemma 8. For each $i$, we can use a single layer to make $\phi_{l+1,i}$ orthogonal with to $\phi_{l,j}$ for all $j > i$. By putting $k$ such layers in sequence, we can ensure that $\phi_{l+k,i}$ is orthogonal to $\phi_{l+k,j}$ for all $i = 1...k$ and for all $j = i + 1...k$. Thus $\Phi_{l+k}$ is exactly the column-orthogonal matrix in a QR decomposition of $\Phi_l$.

Finally, we observe that $B_l$ is unchanged in both (14) and in the construction of Lemma 8. Thus we verify the assumption at the beginning of the proof that $B_l = B$ for all $l$. This concludes the proof.

**Proof of Corollary 7**

The construction is almost identical to that in the proof of Lemma 6 above. The only difference is that we replace the weight configuration in (14) by

$$W_l^V = \begin{bmatrix} 0_{d\times d} & 0_{d\times k} \\ 0_{k\times d} & I_{k\times k} \end{bmatrix}, \qquad W_l^{Q\top} W_l^K = \begin{bmatrix} I_{d\times d} & 0_{d\times k} \\ 0_{k\times d} & 0_{k\times k} \end{bmatrix}, \qquad W_l^R = \begin{bmatrix} I_{d\times d} & 0_{d\times k} \\ 0_{k\times d} & (\mu - 1)I_{k\times k} \end{bmatrix},$$

where the change is highlighted in red. Under this, we verify that $\hat{\Phi}_{l+1} = (\mu I - \mathcal{L})\Phi_l$.

The remainder of the proof, including construction for the $QR(\Phi)$ operation, are unchanged from Lemma 6.

## 12 LEMMAS AND PROOFS FOR SECTION 5

**Lemma 9 (Constructions under (7))** *The constructions in Lemmas 1, 2, 3 and 6 can be realized within the constrained dynamics (7).*

**Proof of Lemma 9**

In general, we verify that (7) is equivalent to (3) with weights satisfying the following form:

$$W_l^V = \begin{bmatrix} \alpha_l^V I_{d\times d} & 0_{d\times 2k} \\ 0_{2k\times d} & W_l^{V,\Phi} \end{bmatrix}, W_l^Q = \begin{bmatrix} \alpha_l^Q I_{d\times d} & 0_{d\times 2k} \\ 0_{2k\times d} & W_l^{Q,\Phi} \end{bmatrix}, W_l^K = \begin{bmatrix} \alpha_l^K I_{d\times d} & 0_{d\times 2k} \\ 0_{2k\times d} & W_l^{K,\Phi} \end{bmatrix}.$$

$$W_l^R = \begin{bmatrix} \alpha_l^R I_{d\times d} & 0_{d\times 2k} \\ 0_{2k\times d} & W_l^{R,\Phi} \end{bmatrix}.$$

Below, we state the weight configurations for (7) which will recover the constructions in each of the stated lemmas. Note that there is a small change in notation: $\Phi_l$ as defined in Section 5 corresponds to $[\Lambda_l; \Phi_l]$ from the proofs of Lemmas 1, 2 and 3.

We leave the simple verification of this equivalence to the reader.

The construction in Lemma 1 is equivalent to (7) with weight configuration $\alpha_l^V = 0, \alpha_l^Q = \alpha_l^K = 1$, $\alpha_l^R = 0, W_l^{V,\Phi} = \begin{bmatrix} 0_{k\times k} & 0_{k\times k} \\ -\delta I_{k\times k} & 0_{k\times k} \end{bmatrix}, W_l^{Q,\Phi} = W_l^{K,\Phi} = 0, W_l^R = \begin{bmatrix} 0_{k\times k} & 0_{k\times k} \\ \delta I_{k\times k} & 0_{k\times k} \end{bmatrix}$.

The construction in Lemma 2 is equivalent to (7) with weight configuration $\alpha_l^V = 0, \alpha_l^Q = \alpha_l^K = 1$, $\alpha_l^R = 0, W_l^{V,\Phi} = \begin{bmatrix} -\frac{1}{\lambda_{max}} I_{k\times k} & 0_{k\times k} \\ 0_{k\times k} & 0_{k\times k} \end{bmatrix}, W_l^{Q,\Phi} = W_l^{K,\Phi} = 0, W_l^R = \begin{bmatrix} 0_{k\times k} & 0_{k\times k} \\ \frac{1}{\lambda_{max}} I_{k\times k} & 0_{k\times k} \end{bmatrix}$.

The construction in Lemma 3 is equivalent to (7) with weight configuration $\alpha_l^V = 0$, $\alpha_l^Q = \alpha_l^K = 1$, $\alpha_l^R = 0$, $W_l^{V,\Phi} = \begin{bmatrix} I_{k\times k} & 0_{k\times k} \\ 0_{k\times k} & 0_{k\times k} \end{bmatrix}$, $W_l^{Q,\Phi} = W_l^{K,\Phi} = 0$, $W_l^R = \begin{bmatrix} -I_{k\times k} & 0_{k\times k} \\ \frac{(-s)^l}{l!} I_{k\times k} & 0_{k\times k} \end{bmatrix}$, where $s$ is the temperature parameter.

The construction in Lemma 6 is equivalent to (7) with the following weight configurations:

1. To implement the first step inside the loop of Algorithm 1, let $\alpha_l^V = 0$, $\alpha_l^Q = \alpha_l^K = 1$, $\alpha_l^R = 0$, $W_l^{V,\Phi} = \begin{bmatrix} 0_{k\times k} & 0_{k\times k} \\ 0_{k\times k} & I_{k\times k} \end{bmatrix}$, $W_l^{Q,\Phi} = W_l^{K,\Phi} = 0$, $W_l^R = \begin{bmatrix} 0_{k\times k} & 0_{k\times k} \\ 0_{k\times k} & -I_{k\times k} \end{bmatrix}$.

2. To implement the QR decomposition step inside the loop of Algorithm 1, we will need to an equivalent construction as in Lemma 8. Let $\alpha_l^V = 0$, $\alpha_l^Q = \alpha_l^K = 1$, $\alpha_l^R = 0$, and let $W_l^{V,\Phi} = \begin{bmatrix} 0_{k\times k} & 0_{k\times k} \\ 0_{k\times k} & A \end{bmatrix}$, $W_l^{Q,\Phi\top} W_l^{K,\Phi} = \begin{bmatrix} 0_{k\times k} & 0_{k\times k} \\ 0_{k\times k} & H \end{bmatrix}$, $W_l^R = 0$, where $A$ and $H$ are as defined in the proof of Lemma 8.

**Lemma 10 (Invariance and Equivariance)** *Let $\mathsf{TF}_l^B$ and $\mathsf{TF}_l^\Phi$ be as defined in Section 5. Let $U \in \mathbb{R}^{d\times d}$ be any permutation matrix. Then for all layers $l$,*

$$\mathsf{TF}_l^B(BU, \Phi) = U^\top \mathsf{TF}_l^B(B, \Phi)$$
$$\mathsf{TF}_l^\Phi(BU, \Phi) = \mathsf{TF}_l^\Phi(B, \Phi) \qquad (15)$$

**Proof**
We will prove these two claims by induction simultaneously. Recall from (7) that

$$B_{l+1}^\top = (1 + \alpha_{R,l})B_l^\top + \alpha_{V,l} B_l^\top \left( \alpha_{Q,l}\alpha_{K,l} B_l B_l^\top + \Phi_l W_l^{Q,\Phi} W_l^{K,\Phi} \Phi_l^\top \right)$$
$$\Phi_{l+1}^\top = \left( I + W_l^{R,\Phi} \right)\Phi_l^\top + W_l^{V,\Phi}\Phi_l^\top \left( \alpha_{Q,l}\alpha_{K,l} B_l B_l^\top + \Phi_l W_l^{Q,\Phi} W_l^{K,\Phi} \Phi_l^\top \right).$$

Recall from the definition that $\mathsf{TF}_0^B(B, \Phi) := B_0 := B$ and $\mathsf{TF}_l^\Phi(B, \Phi) := \Phi_0 := \Phi$. Thus (15) holds by definition. Now suppose (15) holds for some $l$. By (7),

$$\mathsf{TF}_{l+1}^B(BU, \Phi)$$
$$= (1 + \alpha_{R,l})\mathsf{TF}_l^B(BU, \Phi)$$
$$\quad + \alpha_{V,l}\mathsf{TF}_l^B(BU, \Phi)\left( \alpha_{Q,l}\alpha_{K,l}\mathsf{TF}_l^B(BU, \Phi)^\top \mathsf{TF}_l^B(BU, \Phi) \right)$$
$$\quad + \alpha_{V,l}\mathsf{TF}_l^B(BU, \Phi)\left( \mathsf{TF}_l^\Phi(BU, \Phi)^\top W_l^{Q,\Phi} W_l^{K,\Phi}\mathsf{TF}_l^\Phi(BU, \Phi) \right)$$
$$= (1 + \alpha_{R,l})U^\top \mathsf{TF}_l^B(B, \Phi)$$
$$\quad + \alpha_{V,l}U^\top \mathsf{TF}_l^B(B, \Phi)\left( \alpha_{Q,l}\alpha_{K,l}\mathsf{TF}_l^B(B, \Phi)^\top \mathsf{TF}_l^B(B, \Phi) \right)$$
$$\quad + \alpha_{V,l}U^\top \mathsf{TF}_l^B(B, \Phi)\left( \mathsf{TF}_l^\Phi(B, \Phi)^\top W_l^{Q,\Phi} W_l^{K,\Phi}\mathsf{TF}_l^\Phi(B, \Phi) \right)$$
$$= U^\top \mathsf{TF}_{l+1}^B(B, \Phi),$$

where we use the fact that $UU^\top = I_{d\times d}$. By similar steps as above, we also verify that

$$\mathsf{TF}_{l+1}^\Phi(BU, \Phi) = \mathsf{TF}_{l+1}^\Phi(B, \Phi).$$

This concludes the proof.

## 13 DETAILS FOR SYNTHETIC EXPERIMENTS

We consider two ways of sampling random graphs:

1. Fully Connected (FC) graphs: $n = 10$ nodes and $d = 45$ edges.

2. Circular Skip Links (CSL) graphs: $n = 10$ nodes and $d = 20$ edges. The skip-length is sampled uniformly at random from $\{2, 4, 6, 8\}$. See (Dwivedi et al., 2023) for a detailed definition of CSL graphs.

For both FC and CSL graphs, the resistance of an edge $e$ is $r(e) = e^{u(e)}$, where $u(e)$ is independently sampled uniformly from $[-2, 2]$.

# 14 DETAILS FOR MOLECULAR REGRESSION EXPERIMENT

Below, we provide various details for the molecular regression experiment in Section 6.

## 14.1 DATASET DETAILS

For QM9, the training set size is 20,000, the validation set size is 2000, the test set size is 100,000. The training/validation set are subsampled from the full training/validation set. The average number of nodes is 8.79, and the average number of edges is 18.8.

For ZINC, the training set size is 20,000, the validation set size is 2000, the test set size is 24,445. The training/validation set are subsampled from the full training/validation set. The average number of nodes is 23.16, and the average number of edges is 39.83.

## 14.2 ARCHITECTURE FOR MOLECULAR REGRESSION EXPERIMENT

The linear Transformer we use for the experiment in Section 6 is described in (16) below:

$$B_{l+1}^\top = (1 + \alpha_{R,l})B_l^\top + \alpha_{V,l}B_l^\top\left(\beta_{l,1}\alpha_{Q,l}\alpha_{K,l}D_l^{-1/2}B_lB_l^\top D_l^{-1/2} + \beta_{l,2}\Phi_lW_l^{Q,\Phi}W_l^{K,\Phi}\Phi_l^\top\right)$$

$$\Phi_{l+1}^\top = \left(I + W_l^{R,\Phi}\right)\Phi_l^\top + W_l^{V,\Phi}\Phi_l^\top\left(\beta_{l,3}\alpha_{Q,l}\alpha_{K,l}D_l^{-1/2}B_lB_l^\top D_l^{-1/2} + \beta_{l,4}\Phi_lW_l^{Q,\Phi}W_l^{K,\Phi}\Phi_l^\top\right)$$

$$B_{l+1} \leftarrow B_{l+1}/\|B_{l+1}\|_F$$

$$\phi_{l+1,i} \leftarrow \phi_{l+1,i}/\|\phi_{l+1,i}\|_2 \qquad \text{(for i=1...k)} \tag{16}$$

It is similar to the memory-efficient Transformer architecture (7) from Section 5, with a number of modifications which we explain below. Let the input to the transformer be $Z_0 \in \mathbb{R}^{d+k}$, where $d$ denotes the number of edges, and $k$ is the dimension of learned features.

1. **Scaling by $D^{-1}$**: The GT in Dwivedi & Bresson (2020) uses eigenvectors of the *normalized Laplacian* $\bar{\mathcal{L}} := D^{-1/2}\mathcal{L}D^{-1/2}$, where $D$ is the diagonal matrix whose $i^{th}$ diagonal entry is given by $[D]_{ii} := |\mathcal{L}_{ii}| = \sum_{j=1}^d |B_{ij}|$. To be consistent with this setup, we modify the dynamics in (7) to add the scaling by $D_l^{-1/2}$ on the part of the self-similarity matrix involving $B_l$, highlighted in red in (16), where $D_l$ is the diagonal matrix with the $i^{th}$ diagonal entry given by

$$[D_l]_{ii} := \sum_{j=1}^d |B_l|_{ij}.$$

With this additional scaling in the dynamics, the same weight constructions in all the lemmas in this paper will compute the corresponding quantities for the normalized Laplacian $\bar{\mathcal{L}}$ instead. For instance, with the $D_l^{-1/2}$ scaling, the construction in Lemma 1 computes $\bar{\mathcal{L}}^\dagger$, and the construction in Lemma 6 computes the top-$k$ eigenvectors of $\bar{\mathcal{L}}$. This can be verified using the following two facts: First, in all our constructions, $\Phi_l$ interacts with $B_l$ only via the self-similarity matrix $B_lB_l^\top$. Second, $D^{-1/2}BB^\top D^{-1/2} = \bar{\mathcal{L}}$.

2. **Independently scaled similarity matrix:** For each layer $l$, we introduce additional scalar parameters $\beta_{l,1}, \beta_{l,2}, \beta_{l,3}, \beta_{l,4}$. One layer of (16) is more expressive than one layer of (7) but less expressive than two layers of (7) (ignoring the difference due to $D^{-1/2}$).

3. **Diagonal constraints on $W$:** We constrain $W^{V,\Phi}, W^{Q,\Phi}, W^{K,\Phi}$ to be diagonal, and $W^{R,\Phi}$ to be a scaling of identity.

4. **Weight sharing:** Layers $3l, 3l + 1, 3l + 2$ share the same parameters, for all integers $l$. Phrased another way, each layer is looped 3 times.

5. **Per-layer Scaling:** after each layer, we scale $B_l$ to have Frobenius norm 1, and we scale each column of $\Phi_l$ to have Euclidean norm 1.

Items 2-4 are useful for reducing the parameter count and improving generalization. Item 5 makes training feasible for deeper layers. The construction in Lemma 4 for subspace iteration can still be realised under the changes in items 2-5. Under the change in item 1, the construction in Lemma 4 will find eigenvectors of $\bar{\mathcal{L}}$ instead of $\mathcal{L}$.

### 14.3   EXPERIMENT DETAILS

The GT we use has 128 hidden dimensions, 8 heads, and 4 layers. The position encoding dimension is 3 for QM9, and 6 for ZINC.

The linear Transformer we use contain $L = 9$ layers, with parameters shared every 3 layers, as described in Section 14.2. The dimension of $\Phi_l$ is 8. We use a `linear` map, with parameters $M$, to project $\Phi_L$ down to the PE dimension (PE_dim=3 for QM9 and PE_dim=6 for ZINC). This output is then passed to the GNN in the same way as the LapPE. To clarify the distinction among the three models, we provide pseudo-code for **Graph Transformer**, **Graph Transformer + LapPE** and **Graph Transformer + linear Transformer** below:

Let `GT` denote the graph transformer from (Dwivedi & Bresson, 2020). Let $G$ denote the collection of basic graph information, including edge list, edge features, and node features. Let `LapPE(G)` return the Laplacian eigenvector positional encoding for $G$. Let $B$ be the incidence matrix of $G$. Let `LT` denote the linear Transformer, which takes $B$ as input ($\Phi_0$ can be viewed as internal parameters of `LT`)

Then the predictions for each model in Table 2 are given by

| | |
|---|---|
| **Graph Transformer** | `GT`$(G, \texttt{None})$ |
| **Graph Transformer + LapPE** | `GT`$(G, \texttt{LapPE}(G))$ |
| **Graph Transformer + linear Transformer** | `GT`$(G, \texttt{linear}(M, \texttt{LT}(B)))$    (17) |

The trainable parameters are given by

$$\{\text{all the parameters of } \texttt{GT}\} + \{\text{all the parameters of } \texttt{LT}\}$$
$$= \{\text{all the parameters of } \texttt{GT}\} + \{M, \Phi_0, (\alpha\text{'s}, \beta\text{'s}, W\text{'s from (16)})\}.$$

Before training on the actual regression task, we pretrain the linear Transformer to return the top PE_dim Laplacian eigenvectors. This gives a good initialization, and improves the optimization landscape.

We train using AdamW. Graph Transformer parameters are updated with initial 0.001 lr. Linear Transformer parameters are updated with initial 0.01 lr. For ZINC, we train for 2000 epochs with lr halved every 800 epochs. For QM9, we train for 1000 epochs with lr halved ever 400 epochs. The means and standard deviations in Table 2 are each computed over 4 independent seeded runs.

## 15 EXPERIMENTS FOR EFFICIENT IMPLEMENTATION

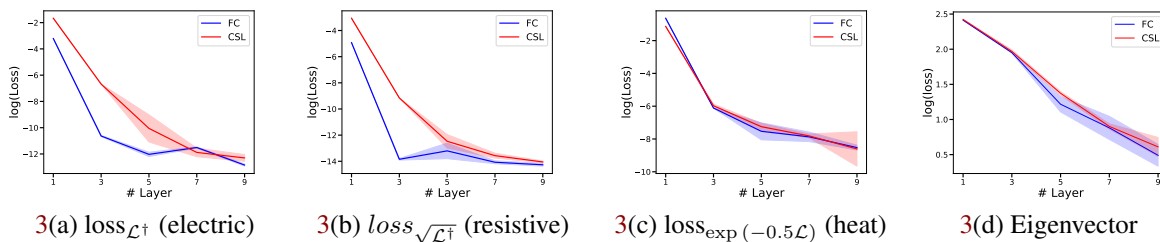

    3(a) $\mathrm{loss}_{\mathcal{L}^\dagger}$ (electric)      3(b) $loss_{\sqrt{\mathcal{L}^\dagger}}$ (resistive)      3(c) $\mathrm{loss}_{\exp(-0.5\mathcal{L})}$ (heat)      3(d) Eigenvector

Figure 3: Plot of loss against number of layers for the 4 problems. Figures {3(a), 3(b), 3(c), 3(d)} correspond to Figures {1(a),1(b),1(c),2(d)} respectively. The experiment setup of each corresponding pair of plots are identical, **except for the architecture used**: all plots in Figure 3 are made using the efficient implementation described in Section 5.

