# OpenReview forum: "Graph Transformers Dream of Electric Flow"
_ICLR.cc/2025/Conference — ICLR 2025 Poster_

### Official Review · Reviewer_PYgD · 2024-11-01

**Soundness:** 3
**Presentation:** 3
**Contribution:** 2
**Rating:** 6
**Confidence:** 2

**Summary:**

The papers show theoretically and empirically that the linear transformer can approximate the hypothesis function space of several graph problem (electric flow and Laplacian eigenvector decomposition). The authors also conduct a real-world experiment on molecular regression, in which positional encoding outputs by transformers are adopted.

**Strengths:**

* The organisation of the paper is clear, it's easy to understand the main contribution of the paper. And mathematically, it's generally well-written.
* The idea of solving the traditional graph problem with a Transformer is novel.
* For the proposed lemmas, the paper provides the relevant experiments which are quite nice.

**Weaknesses:**

* My main concern is that since the neural network is a universe approximator, it's not surprising that it (in this paper, even it concerns linear Transformer, which is more like in a polynomial function) can solve approximate the hypothesis function space of several graph problems. I wonder how the lemmas shown in the paper can provide us with more useful information.
* The paper lacks a conclusion section.
* The real-world experiment concerns only the Laplacian eigenvector decomposition, it would be nice to have another for the electric flow.
* See the questions.

**Questions:**

* While the paper concerns two graph problems: electric flow and Laplacian eigenvector decomposition, why the title mentions only the first one?

---

> ### Author Response · Authors · 2024-11-16
> **Response to reviewer PYgD**
>
> Thank you for your constructive comments and feedback. We address your questions below:
>
> > My main concern is that since the neural network is a universe approximator, it's not surprising that it (in this paper, even it concerns linear Transformer, which is more like in a polynomial function) can solve approximate the hypothesis function space of several graph problems. I wonder how the lemmas shown in the paper can provide us with more useful information.
>
> A number of works have indeed claimed that Transformers have high expressive power in theory, and given enough parameters, can simulate a large class of functions.
>
> However, there is a catch to these "universal approximation results" -- the constructions are usually **very expensive**; in some cases, the number of parameters needed **can grow exponentially in dimension and 1/ (target accuracy)** (see e.g. [Corollary 4.1, (Chong 2020)](https://openreview.net/pdf?id=rkevSgrtPr)). Additionally, the constructions are often **highly complex, and may thus require prohibitive numbers of training samples**.
>
> Similar universal-approximation results exist for sufficiently-wide two-layer relu networks. Therefore, these universal approximation results may not explain **why Transformers work in practice**; it is also unclear how the universal approximation results help us design better architectures.
>
> In contrast, our result requires only a modest amount of memory. To handle graphs with up to $n$ nodes and $m$ edges, our construction uses $O(n^2 + m^2)$ parameters per layer. The efficient implementation of Section 5 further has a per-layer parameter-count of only **O(#query^2)**, independent of $n$ and $m$. Furthermore, the error $\epsilon$ **shrinks exponentially fast with number of layers** (in the case of Lemma 4, doubly exponentially). The experiment in Section 6 demonstrates that **the architecture suggested by our theory can indeed outperform a baseline Graph Transformer on a real-world task, using only 488 parameters**.
>
> Finally, though a rigorous analysis of the loss landscape is beyond the scope of this paper, we note that our constructions are simple -- $W^Q, W^K, W^V$ are usually just blocks of zero or identity; this simplicity makes it hopeful to establish optimality/convergence/learning guarantees in future work.
>
> > The paper lacks a conclusion section.
>
> Thank you for the suggestion. We will try to add a conclusion section in the next draft.
>
> > The real-world experiment concerns only the Laplacian eigenvector decomposition, it would be nice to have another for the electric flow.
>
> This is an excellent suggestion. In the future, we plan to study architectures which use resistive embeddings, and see if there is a benefit in replacing the resistive embedding by a linear Transformer.
>
> We do note that the PE learned by the linear Transformer in Table 2 (molecular regression) is **no longer LapPE by the end of training**. It is possible that the learned embedding contains mixtures of resistive/electrical/eigenvector embeddings, but further analysis is needed.
>
> > While the paper concerns two graph problems: electric flow and Laplacian eigenvector decomposition, why the title mentions only the first one?
>
> This is a sharp observation. The title is chosen for two reasons:
> 1. We chose to emphasize electric flow because it is the graph analog of linear-regression, as both involve solving a linear system, or equivalently minimizing a quadratic objective. Just like quadratic minimization is arguably the prototypical optimization problem, **graph electric flow is arguably the prototypical problem in spectral graph theory**; practically, it also serves as a building block in fast solvers for many other graph problems.
> 2. The Transformer constructions for electric flow have a particularly nice interpretations, i.e. gradient descent (Lemma 1) and iteratively preconditioned gradient descent (Lemma 4) (see e.g. [Peng Spielman 2013](https://arxiv.org/pdf/1311.3286)). As noted in our paper on line 118, these two constructions also have surprising connections to a recent line of work on the In-context Learning capability of Transformers.

---

### Official Review · Reviewer_BwGA · 2024-11-03

**Soundness:** 3
**Presentation:** 2
**Contribution:** 2
**Rating:** 6
**Confidence:** 4

**Summary:**

The paper shows that Linear Transformers (and their variant) using the incidence matrix can implement canonical problems, such as electric flow and subspace iteration. While the contribution is mainly theoretical, the authors also empirically validate their findings, and further show that on existing molecular datasets the linear transformer is capable of learning better performing positional encodings than standard laplacian eigenvectors.

**Strengths:**

The results are interesting. I particularly appreciated the structure of the paper, in which experimental results are shown after their corresponding theoretical claims instead of having them all the end, which highlights the connection between the presented theory and the experimental results.

**Weaknesses:**

W1. Generally speaking, I think the impact of the paper tends to be a bit limited. This is especially true because the considered architecture (linear Transformers), and in particular its variant which includes an L2 normalization, is not widely used. The main implication I can see is to use the variant of the linear transformer in place of existing predefined positional encodings, and therefore as an additional component of a bigger architecture.


W2. I find a bit confusing that the authors claim that the input to the Transformer is only the graph incidence matrix, without any other positional encoding (Lines 11-13), while for instance for the subspace iteration $\Phi_0$ is required to be some arbitrary column-orthogonal matrix. I think -- depending on the graph -- $\Phi_0$ might need to be a positional encoding to ensure all columns are orthogonal (note that if $\Phi_0$ is a random matrix then it is a positional encoding [Srinivasan and Ribeiro, 2020]). I think the authors need to clarify this point and adjust accordingly in the entire paper.


Srinivasan and Ribeiro, 2020. On the Equivalence between Positional Node Embeddings and Structural Graph Representations.

**Questions:**

Q1. Can you elaborate on the practical implications of your findings (see W1)?

Q2. I think the eigenvectors you converge to have sign/basis ambiguities derived from the choice of $\Phi_0$. How do you deal with these ambiguities? Do you perform sign flipping at every forward pass?

Q3. I think denoting by $d$ the number of edges (Line 128) is a bit confusing, because $d$ is usually the embedding dimension. Also, in Equation 1, since $j= 1, \ldots, d$ is the first index of $B$, I think $B \in \mathbb{R}^{d \times n }$, but you wrote $B \in \mathbb{R}^{n \times d }$ (Line 132). Please clarify.

---

> ### Author Response · Authors · 2024-11-16
> **Response to reviewer BwGA**
>
> Thank you for your constructive comments and feedback. We address your questions below:
>
> > W1. Generally speaking, I think the impact of the paper tends to be a bit limited. This is especially true because the considered architecture (linear Transformers), and in particular its variant which includes an L2 normalization, is not widely used. The main implication I can see is to use the variant of the linear transformer in place of existing predefined positional encodings, and therefore as an additional component of a bigger architecture.
>
> > Q1. Can you elaborate on the practical implications of your findings (see W1)?
>
> **On practical implications:**
> 1. One practical implication, as already noted by reviewer BWGA, is showing the potential of linear Transformers as a component for learning a positional encoding that is better than pre-defined PEs.
> 2. A more general take-away from our paper is in showing how the attention module is **surprisingly well-suited for computing a range of interesting graph properties**. The 3rd-order polynomial in $Z$, given by $W_V Z Z^\top W_Q^\top W_K Z$, is exactly what one needs to implement algorithms like gradient descent for electric energy (Lemma 1), or even the "iteratively preconditioned gradient descent" with $\log\log(1/\epsilon)$ convergence (Lemma 4), related to iteratively-preconditioned gradient descent.
> We believe that this understanding gives **useful insights in the design of more efficient graph transformer architectures**. For instance, both the removal of softmax, and the enforcement of sparse structure on $W_Q, W_K, W_V$ can significantly reduce memory and speed-up computation (Section 5).
>
> It might be interesting to note that the $L_2$ normalization in our paper is applied to each "feature dimension"; the direction of normalization is the same as *BatchNorm*. [(Dwivedi and Bresson 2020)](https://arxiv.org/pdf/2012.09699) reported that BatchNorm gives significantly better test performance than LayerNorms in their Graph Transformer experiments.
>
>
> > W2. I find a bit confusing that the authors claim that the input to the Transformer is only the graph incidence matrix, without any other positional encoding (Lines 11-13), while for instance for the subspace iteration $\Phi_0$ is required to be some arbitrary column-orthogonal matrix. I think -- depending on the graph -- $\Phi_0$ might need to be a positional encoding to ensure all columns are orthogonal (note that if $\Phi_0$ is a random matrix then it is a positional encoding [(Srinivasan and Ribeiro, 2020)](https://arxiv.org/pdf/1910.00452)). I think the authors need to clarify this point and adjust accordingly in the entire paper.
>
> This is a good point, and we thank the reviewer for bringing this up. Our original statement on line 11-13 is a little imprecise. The following is a precise statement:
>
> *The Transformer has **access to information on the input graph $G$ only via $G$'s incidence matrix**; no other explicit positional encoding information for $G$ is provided in the input.*
>
> Regarding $\Phi_0$: we believe that $\Phi_0$ should not be considered as positional encoding, because **$\Phi_0$ does not depend on the input graph.**
>
> 1. For Electric Flow (Lemma 1, 4), Resistive Embedding (Lemma 2), $\Psi$ (analog of $\Phi_0$) acts as an **arbitrary demand**: for a fixed input graph $G$ with Laplacian $\mathcal{L}$ and incidence matrix $B$, and for **any $\Psi$**, the Transformer must outpput $(\mathcal{L}^{\dagger} \Psi)^\top$.
> 2. For subspace iteration, in our experiments $\Phi_0$ is a learnable but fixed matrix. I.e. **for any input graph $G$, $\Phi_0$ is the same constant matrix**, and thus cannot encode the information specific to $G$.
> 3. On line 386 we do state that $\Phi_0$ is a column-orthogonal matrix. Our analysis really only needs the weaker condition that $\Phi_0$ is *non-degenerate* (otherwise power iteration will fail). We state the stronger column-orthogonal condition to avoid theoretical numerical issues due to highly-aligned singular vectors. This should never occur in practice if $\Phi_0$ is randomly sampled from, say, the Gaussian. We will add a clarification to the next draft.
>
> We thank the reviewer for pointing us to [(Srinivasan and Ribeiro, 2020)](https://arxiv.org/pdf/1910.00452)). The perspective on random node embedding (with distributional G-equivariance) has some very interesting connections to our results, and we will add a reference in the next draft. However, we believe that $\Phi_0$ does not fit Definition 12 of [(Srinivasan and Ribeiro, 2020)](https://arxiv.org/pdf/1910.00452)), because as noted above, $\Phi_0$ (or its distribution, if sampled randomly) does not depend on the input graph $G$.
>
> *Minor note: of course, if $B$ is a positional encoding, then $[B,\Phi]$ is also a positional encoding, but $[B,\Phi]$ gives no more information about the graph $G$ than $B$ alone.*
>
> (continued in next comment)

---

> > ### Author Response · Authors · 2024-11-16
> > **Response to reviewer BwGA (cont)**
> >
> > > Q2. I think the eigenvectors you converge to have sign/basis ambiguities derived from the choice of $\Phi_0$. How do you deal with these ambiguities? Do you perform sign flipping at every forward pass?
> >
> > This is a great question. When training the linear Transformer to output the eigenvector (both for the synthetic experiment in Figure 2, as well as for pretraining the linear Transformer for the molecular regression task in Table 2), the **training loss is invariant to sign-flips**. To be precise, let $v_2$ denote the second-smallest Laplacian eigenvector, and let $\phi_2$ denote the vector from the second row of the Transformer's output. Then the loss is given by
> > $$\text{loss}_2 = \min \{\|\phi_2-v_2\|_2, \|\phi_2+v_2\|_2\}.$$
> > (this is defined on line 426.)
> >
> > Since $\Phi_0$ is a learnable parameter (see discussion on line 425), given any fixed graph input, the Transformer will likely output eigenvectors with a arbitrary-but-fixed sign orientation. We do not perform random sign-flipping because the Transformer seems to work well even without sign-flipping on the molecular regression task. (We do perform sign-flipping on the input LapPE for the GT+LapPE baseline of Table 2, as it does significantly decrease the loss there.)
> >
> > Finally, we note that the construction of Lemma 6, based on subspace iteration, is *in theory* sign-equivariant to $\Phi_0$, i.e. flipping sign on the $i^{th}$ row of $\Phi_0^\top$ will also flip sign on the Transformer's prediction for the $i^{th}$ eigenvector. We conjecture that this equivariance will be learned if $\Phi_0$ were not a learnable parameter, but is instead randomly chosen for each input graph. However, we yet to test this experimentally.
> >
> >
> > > Q3. I think denoting by $d$ the number of edges (Line 128) is a bit confusing, because is usually the embedding dimension. Also, in Equation 1, since $j=1,...,d$ is the first index of $B$, I think $B\in \mathbb{R}^{d\times n}$, but you wrote $B\in \mathbb{R}^{n\times d}$ (Line 132). Please clarify.
> >
> > This is an excellent catch, thank you for your careful read of the paper. The typo is in equation $(1)$ on line 134-137: the left hand side of $(1)$ should read $B_{i,j}$ instead of $B_{j,i}$. However $B\in \mathbb{R}^{n\times d}$ on line 132 is indeed correct, and is consistent with the usage of $B$ in the rest of the paper. We will fix this typo in the next draft.
> >
> > On the choice of $d$: the embedding dimension of our constructions are bounded by #edge + 2* #vertex, where #edge is usually the dominant term. This is why we chose to use $d$ for the number of edges. This can indeed be confusing, and we will consider a different letter (such as $m$) for the number of edges in the next draft.

---

> > > ### Comment · Reviewer_BwGA · 2024-11-19
> > >
> > > I would like to thank the authors for their reply.
> > >
> > >  I disagree that $\phi_0$ does not fit Definition 12 in Srinivasan and Ribeiro, 2020. First of all $\phi_0$ depends on the graph because it depends on the number of its nodes. Then, the definition is satisfied by simply considering that  $\phi_0$ is obtained from a probability distribution $p$ that does not consider the actual adjacency matrix, which is allowed by definition 12 (since it  doesn't require the positional encoding to be different for different graphs). For this reason random node features are indeed (special cases of) positional encoding.
> > >
> > > Moreover, if one of the practical implications of the paper is to learn positional encodings, then the authors are missing comparisons with existing methods addressing the same problem, e.g., [1, 2, 3]. Also, to really claim that this approach can learn positional encoding better than predefined approaches, I think you need to include a comparison with other predefined positional encodings including random walk positional encodings [4] .
> > >
> > > [1] GRPE: Relative Positional Encoding for Graph Transformer.
> > >
> > > [2] GraphiT: Encoding graph structure in transformers.
> > >
> > > [3] Graph Positional Encoding via Random Feature Propagation.
> > >
> > > [4]  Graph neural networks with learnable structural and positional representations.

---

> ### Author Response · Authors · 2024-11-20
> **Response to follow-up comment by reviewer BwGA**
>
> We sincerely thank reviewer BwGA for the follow-up comments and illuminating discussion, as well as for pointing out a number of relevant papers. We address the points below:
>
> > I disagree that $\phi_0$ does not fit Definition 12 in Srinivasan and Ribeiro, 2020. First of all $\phi_0$ depends on the graph because it depends on the number of its nodes. Then, the definition is satisfied by simply considering that $\phi_0$ is obtained from a probability distribution $p$ that does not consider the actual adjacency matrix, which is allowed by definition 12 (since it doesn't require the positional encoding to be different for different graphs). For this reason random node features are indeed (special cases of) positional encoding.
>
> We quote Definition 12 of [(Srinivasan and Ribeiro, 2020)](https://arxiv.org/pdf/1910.00452) below:
> > **Definition 12**
> The node embeddings of a graph $G=(A,X)$ are defined as joint samples of random variables $(Z_i)_{i\in V}\vert A,X \sim p(\cdot | A,X), Z_i \in \mathbb{R}^d,d\geq 1$, where **$p(\cdot |A,X)$ is a $G$-equivariant probability distribution on $A$ and $X$, that is, $\pi(p(\cdot |A,X)) = p(\cdot|\pi(A), \pi(X))$ for any permutation $\pi\in \Pi_n$**. (emphasis ours)
>
> In the above, $A$ is the $n\times n$ adjacency matrix, and $X$ is the $n\times k$ node-feature matrix. Crucially, the distribution $p(\cdot |A,X)$ is required to be node-permutation equivariant, i.e. if we permute $A$ and $X$ by $\pi$, the distribution of $Z$ must be likewise permuted. On a general $n$-node graph, $G$-equivariance **cannot be satisfied** if $\Phi_0$ does not depend on $G$. In the language of Definition 12, let $p(\cdot |A,X)$ denote the distribution of $\Phi_0$ (which is a point mass of probability 1 since $\Phi_0$ is deterministic). Under our choice of $\Phi_0$,
> $$p(\cdot |A,X) = p(\cdot|\pi(A), \pi(X))$$ but to satisfy $G$-equivariance, we need
> $$\pi(p(\cdot |A,X)) = p(\cdot|\pi(A), \pi(X)).$$ In other words, ***$G$-equivariance does require $\Phi_0$ to be different (specifically $\pi$-permuted) for $G$ and $\pi(G)$.***
>
> Despite the above, we would like to **emphasize again that we do not wish to dispute what gets called a PE**, as there are many accepted definitions, beyond Definition 12 above. Please let us know if we addressed your concern with this explanation, or if you believe we are still missing anything major?
>
> (continued in next comment due to word limit)

---

> > ### Author Response · Authors · 2024-11-20
> > **Response to follow-up comment by reviewer BwGA (cont)**
> >
> > > ... to really claim that this approach can learn positional encoding better than predefined approaches, I think you need to include a comparison with other predefined positional encodings including random walk positional encodings [4].
> > [4] Graph neural networks with learnable structural and positional representations.
> >
> > We thank the reviewer for the suggestion. In our **specific setup** with GT for molecular regression, we also tried electric/resistance/heat based positional encodings, but they were **substantially worse** than LapPE, and so were not mentioned. Since LapPE is always used for standard GT in literature, it is likely that LapPE is particularly well suited for the standard GT, so we focused on that as the baseline for comparison.
> >
> > We do acknowledge that the **optimality of a positional encoding choice** depends crucially on the **architecture used**. E.g. resistance distance works poorly for GT but works very well on GD-WL in [Zhang et al 2023](https://arxiv.org/pdf/2301.09505). Another example is the RWPE+GatedGCN combination in [[4]](https://arxiv.org/pdf/2110.07875) that the reviewer has already mentioned.
> >
> > Zooming out a little bit, we would like to mention again that the goal of our molecule experiment was to **demonstrate the advantage of the linear Transformer's advantage over LapPE in a real-world application**. A comprehensive study of how linear Transformer integrates into different architectures to replace the various PEs is definitely very interesting, but is beyond the scope of this work, and merits independent study.
> >
> > > Moreover, if one of the practical implications of the paper is to learn positional encodings, then the authors are missing comparisons with existing methods addressing the same problem, e.g., [1, 2, 3].
> > [1] GRPE: Relative Positional Encoding for Graph Transformer.
> > [2] GraphiT: Encoding graph structure in transformers.
> > [3] Graph Positional Encoding via Random Feature Propagation.
> >
> > We thank the reviewer for the suggestions. We briefly discuss learned PEs on line 108 of the paper in the context of [Ma et el 2023](https://arxiv.org/abs/2305.17589), and will expand the discussion to include [1,2,3]. [1,2] have **architectural innovations** that **go hand-in-hand with their proposed (learned) PEs**. For instance, [[1]](https://arxiv.org/pdf/2201.12787) designs a new attention mechanism that is used to inject their proposed PE. [[2]](https://arxiv.org/pdf/2303.02918) uses the DSS-GNN architecture that is suited to handling parallel sampled trajectories. The best reported loss for ZINC in [[3]](https://arxiv.org/pdf/2106.05667) is (0.202) which is close to GT+LapPE and wprse than LT+GT (see Table 2 in our paper), but this may not be a fair (apple-to-apples) comparison again due to differences in both architecture and  experiment setup.
> >
> > In summary, we believe that these comparisons are extremely important in the future, if we try to establish that linear Transformers can improve upon the SOTA. However, within the current scope of the paper, our primary goal is to establish the advantage of LT over LapPE in a simple controlled (but real-world motivated) setting.

---

> > > ### Author Response · Authors · 2024-11-24
> > >
> > > BwGA
> > > We just updated our submission draft according to your comments. In particular,
> > > 1. We rephrased the sentence in the abstract in line with our discussion on positional encoding.
> > > 2. In Section 1.2, we added references to the papers that came up in our discussions.
> > > 3. We gave a more careful characterization of the condition on $\Phi_0$ on lines 386 and 393.
> > >
> > > We sincerely thank you for your comments and suggestions which helped us greatly in improving the paper.

---

> > > > ### Comment · Reviewer_BwGA · 2024-11-25
> > > >
> > > > Thank you for the reply.
> > > >
> > > > My main concern was related to the practical implications and impact of the paper, which remains only partially addressed due to the lack of empirical comparisons with learned PEs and with predefined positional encodings other than LapPE.  For this reason I will keep my positive score. Nonetheless, since they addressed other clarifying questions I had I have increased my confidence.

---

> > > > > ### Author Response · Authors · 2024-11-25
> > > > >
> > > > > Thank you once again for your review, and for the the many useful discussions. We will definitely try to add more comprehensive experiments.

---

### Official Review · Reviewer_fauP · 2024-11-04

**Soundness:** 3
**Presentation:** 3
**Contribution:** 3
**Rating:** 6
**Confidence:** 4

**Summary:**

The authors introduce a simplified (linear) Transformer architecture (no softmax activation in self-attention layers) and use it to show (theoretically and empirically) that it can implement graph algorithms. In particular they present a series of lemmas providing bounds in approximating the solution to problems like flow and heat kernel computation, finding resistive embeddings and performing eigenvalue decomposition (assuming specific weight matrices). Their input encodings include the incidence matrix representation of the graph and then the approximate solution will land in a subset of columns in the output encodings' matrix,  after passing through a a number L of transformer layers (where L is decided by the target approximation error). A parameter efficient variant is also presented. The linear transformer is tested on a series of small graph instances of the problems (e.g. of the order of 10 nodes and 20 edges) and for learning a positional encoding (PE) for a molecular regression task.

**Strengths:**

- The idea is novel and appealing: using a transformer to compute the solution to a graph problem as part of the latent vectors in its node output representations adds another level of applicability of transformers well beyond language understanding or generation.

- Lemmas are well organized, follow similar themes and the narrative is smooth and clear.

**Weaknesses:**

- Removing the nonlinear softmax terms from standard transformer architecture, facilitates analysis but severely impacts the power of the model.

- Complexity of the approach is prohibitive: it can be O(n^4) and this explains their experimentation with very small synthetic graphs. Parameter efficient implementation is promising, but still the original idea is far from being scalable and thus practically testable beyond a couple of tens' of graph nodes.

**Questions:**

- How would the analysis be impacted if nonlinearity in self-attention was re-introduced? In order to "understand, at a mechanistic level, how the Transformer processes graph-structured data" (lines 32-33), we are expected to remove only non-essential elements of Transformer architecture (and nonlinearity seems to be a critical one). The linearity in the Transformer, means that if its operations are expanded for a number of layers we'll reduce to a simple matrix operator.

---

> ### Author Response · Authors · 2024-11-16
> **Response to reviewer fauP**
>
> Thank you for your constructive comments and feedback. We address your questions below:
>
> > Removing the nonlinear softmax terms from standard transformer architecture, facilitates analysis but severely impacts the power of the model.
>
> > How would the analysis be impacted if nonlinearity in self-attention was re-introduced? In order to "understand, at a mechanistic level, how the Transformer processes graph-structured data" (lines 32-33), we are expected to remove only non-essential elements of Transformer architecture (and nonlinearity seems to be a critical one). The linearity in the Transformer, means that if its operations are expanded for a number of layers we'll reduce to a simple matrix operator.
>
> Thank you for raising this important point; we are glad to elaborate on this.
> 1. The linear attention module Attn(Z) is ***not* a linear function of its input $Z$**. It is in-fact a 3rd order matrix polynomial in $Z$ (i.e. $W_V Z Z^\top W_Q^\top W_K Z$). Therefore, **more layers of linear attention does significantly increase the representative power of a linear Transformer**. This is **in contrast to** architectures like deep linear fully-connected networks, which are actually linear in its input, and whose expressive power *does not* increase with layers. This is clearly seen from the **exponential decrease in loss with layers** shown in all our synthetic experiments.
> Additionally, we highlight that **the expressive power of softmax Transformer is *not* strictly larger than linear Transformers**; rather, linear and softmax attention are good at representing different classes of functions. In 2b) and 2c) below, we list empirical evidence showing how the linear Transformer is in fact better suited to the problems considered in this paper.
> 3. On softmax-activated Transformers:
>     * a) Appendix A.9 of [von Oswald et al 2023](https://arxiv.org/pdf/2212.07677) show that the softmax attention may approximate linear attention up to a linear offset, and thus it may be possible to approximate a linear attention head with *two softmax attention heads*. Consequently, we **believe that our constructions can be approximated by softmax Transformers *given enough heads***, but careful analysis is needed to understand the error of this approximation.
>     * b) The theory and experiments of [Cheng et al 2023](https://arxiv.org/abs/2312.06528) suggest that softmax activation performs suboptimally compared to linear activation on tasks like ICL of linear functions (see e.g. Figure 1a of linked paper). A similar result may hold for the electric flow problem as well because it also involves solving a linear system.
>     * c) We ran additional experiments (see Figures 1 and 2 on page 1 of [additional plots](https://anonymous.4open.science/r/iclr_2025_rebuttal_plots-3C6/iclr_rebuttal_plots.pdf)), comparing the performance of **1-head linear** Transformer, **1-head softmax** Transformer, and **2-head softmax** Transformer on the Electric Flow and Laplacian Eigenvector problems. In general, **1-head linear > 2-head softmax > 1-head softmax** in performance. The gap between linear and softmax is generally quite large, but linear and 2-head softmax achieve similar loss *for electric flow on CSL graphs*.
>
> EDIT 11/18: added plots for synthetic experiment with efficient implementation. Due to character limit, moved last part of response to next comment.
>
> (continued in next comment)

---

> ### Author Response · Authors · 2024-11-18
> **Response to reviewer fauP (cont)**
>
> > Complexity of the approach is prohibitive: it can be O(n^4) and this explains their experimentation with very small synthetic graphs. Parameter efficient implementation is promising, but still the original idea is far from being scalable and thus practically testable beyond a couple of tens' of graph nodes.
>
> This is an important observation, thank you for pointing this out. We also note this on line 468 of the paper.
>
> We agree that the naive constructions require prohibitive memory. For a graph of $n$ nodes and $m$ edges, the parameter count is $O(m^2)$ which can be as large as $O(n^4)$ (though in many practical setting, $m<<n^2$ due to sparsity). We chose to present the naive constructions in most of our lemmas because of their **theoretical simplicity**. This **does not reflect the practical usefulness** of our constructions.
>
> This is because our efficient construction in Section 5 reduces the parameter count to O(#$\text{query}^2$). Remarkably, this is **independent of $n$ and $m$** and thus scale well to very large graphs. **All the constructions in this paper have corresponding efficient constructions which are exactly equivalent**. We provide one example on line 482, and will provide explicit efficient constructions of all other lemmas in the next draft.
>
> We **verify empirically that the efficient implementation works *at least* as well as the naive implementation**:
> 1. The experiment on molecular regression (Table 2) **uses the efficient construction**, and **demonstrates the practicality of the efficient implementation on a complex real-world task**.
> 2. In Figure 3 on page 2 of [additional plots](https://anonymous.4open.science/r/iclr_2025_rebuttal_plots-3C6/iclr_rebuttal_plots.pdf), we plot the loss against number of layers for each of the 4 tasks: electric, resistive, heat, eigenvector. In all plots, **the Transformer architecture is the efficient implementation described in Section 5 of the paper**. Compared to the corresponding plots in the original paper, the **efficient implementation generally achieves equal-or-better losses than the naive implementation**. This is likely because the Transformer parameters under the efficient implementation is easier to optimize.

---

### Official Review · Reviewer_ryVa · 2024-11-04

**Soundness:** 2
**Presentation:** 3
**Contribution:** 2
**Rating:** 5
**Confidence:** 4

**Summary:**

The paper investigates the capabilities of the linear Transformer when applied to graph data, particularly its ability to implement algorithms for core graph tasks without explicit positional encodings. The authors demonstrate that the linear Transformer, which uses the graph incidence matrix as input, can solve canonical problems like electric flow computation and eigenvector decomposition. Key contributions include explicit configurations for weight matrices, error bounds for each task, and empirical results validating theoretical findings on synthetic data.

**Strengths:**

**Originality**: This paper introduces a novel use of a linear Transformer, to perform core graph algorithms like electric flow and eigenvector decomposition without explicit positional encodings.

**Quality**: The paper offers rigorous theoretical analysis with explicit weight constructions and error bounds for each algorithm.

**Clarity**: The paper is well-organized, clearly written, and enjoyable to read.

**Weaknesses:**

1. Since the paper’s contribution is primarily theoretical, providing a proof sketch under the main results would be highly beneficial. Additionally, a more detailed description of the weight matrices would enhance clarity.

2. The theoretical results are not particularly surprising given the use of a linear Transformer. Could these results also apply to GNNs?

3. The practical impact of the proposed approach is unclear, as the empirical results are limited compared to numerous existing works. For instance, the performance on the ZINC dataset is significantly lower than the state-of-the-art, with an error approximately twice as large.

**Questions:**

Please address the points raised in the previous section.

---

> ### Author Response · Authors · 2024-11-16
> **Response to reviewer ryVa**
>
> Thank you for your constructive comments and feedback. We address your questions below:
>
> > 1. Since the paper’s contribution is primarily theoretical, providing a proof sketch under the main results would be highly beneficial. Additionally, a more detailed description of the weight matrices would enhance clarity.
>
> Thank you for your suggestion on organization; we will try to fit a proof sketch in the main body in the next draft. We do note that for all our constructions, we state the explicit weight configurations in the individual proofs (albeit in the appendix).
>
> > 2. The theoretical results are not particularly surprising given the use of a linear Transformer.
>
> We disagree that the theoretical results are not particularly surprising. Though the representation power of a linear Transformer is high, our construction is **very efficient**: For graphs with $n$ nodes and $m$ edges,
> 1. Number of layers is logarithmic in target accuracy $\epsilon$: $L=\log(1/\epsilon)$ (or even $L=\log\log(1/\epsilon)$).
> 2. Dimension: **m+ 2\*#query**, and has exploitable sparsity for sparse graphs.
> 3. Number of parameters per layer: Standard implementation uses $O(n^2+m^2)$ parameters. Efficient implementation uses **O((#query)^2)** parameters, which is very small and is independent of $n,m$. Empirically, the 9-layer Transformer for learning the positional encoding for ZINC only contained 488 parameters (Table 2).
>
> Prior to our work, it was **not known** how the forward pass of a (pure) Transformer can process an input graph and **efficiently** compute various graph properties of interest. The only relevant result we know of is that Transformers are **universal function approximators** (as also noted by reviewer PYgD); the catch is that **the dimension/layers/memory needed are prohibitively large**, and can grow exponentially in dimension or $1/\epsilon$ (see e.g. [Corollary 4.1, (Chong 2020)](https://openreview.net/pdf?id=rkevSgrtPr)).
>
> Our construction's efficiency is exactly because **the linear attention module is uniquely well-suited to implementing various graph algorithms**. For instance, a single 1-head linear attention layer in Lemma 1 exactly implements 1 step of gradient descent wrt electric energy.
>
> > Could these results also apply to GNNs?
>
> This is a very interesting question; it relates to the question of **whether our constructions fundamentally rely on some aspect the Transformer architecture**.
> 1. For electric flow/resistive distance/heat kernel, we believe that *some construction* of common GNN should exist, but these may be significantly *less efficient* than the Transformer implementation.
> 2. Certain other operations may rely crucially on the attention mechanism. Two concrete examples are the constructions in Lemma 4 (multiplicative polynomial expansion) and Lemma 6 (power iteration, as well as QR step). We suspect that the self-attention matrix is necessary for these constructions.
>
> The exact answer probably depends a lot on the specific architecture in question. E.g. SAN and GAT contain the attention mechanism and may behave more similarly to the Transformer in this paper. On the other hand, MPNNs or GCNs, which are quite different from Transformers, may not be able to efficiently implement some of the algorithms.
>
> (continued in next comment)

---

> ### Author Response · Authors · 2024-11-16
> **Response to reviewer ryVa (cont)**
>
> > 3. The practical impact of the proposed approach is unclear, as the empirical results are limited compared to numerous existing works. For instance, the performance on the ZINC dataset is significantly lower than the state-of-the-art, with an error approximately twice as large.
>
> We agree that our performance on ZINC is significantly lower than the state-of-the-art (we also noted this in our paper on line 532). Nonetheless, our experiment demonstrates the following:
> 1. One does not need to explicitly compute PEs such as Laplacian Eigenvector, the Transformer is capable of learning these PEs during training, **given access to the graph via only the incidence matrix**. As far as we are aware, the SOTA models for ZINC all use some form of non-trivial positional encoding (usually LapPE or resistance).
> 2. Compared to the hard-coded PEs, the PE learned by the linear Transformer (LT) can lead to a **significant improvement** over using LapPE.
>
> Note that both LT+GT and LapPE+GT in Table 2 are based on the Graph Transformer (GT) implementation from [(Dwivedi and Bresson 2020)](https://arxiv.org/pdf/2012.09699). The vanilla implementation of GT is out-performed by newer (and more complex) GNNs. Thus our implementation of LT+GT is also unlikely to achieve the SOTA loss. The large **relative improvement** over LapPE+GT does **demonstrate that the linear Transformer can learn a better PE than Laplacian Eigenvector**.
>
> Finally, we wish to briefly explain our choice using the GT implementation from [(Dwivedi and Bresson 2020)](https://arxiv.org/pdf/2012.09699): The primary goal of our experiment was to **study the change in loss by replacing LapPE with a linear Transformer**. The GT from [(Dwivedi and Bresson 2020)](https://arxiv.org/pdf/2012.09699) is particularly suitable because **conceptually, its architectural design is relatively simple and clean**, and involves few specially-designed-modifications over the standard Transformer.

---

### Author Response · Authors · 2024-11-22
**Summary of  Reviews and Our Response**

We once again thank all the reviewers for their comments and feedback, and for the numerous insightful discussions on various aspects of our paper. Below, we summarize the reviews and our responses

We begin with a summary of the strengths of our paper

## **Strengths**

1. Using a Transformer to solve a graph problem is novel, and adds a level of applicability of Transformers beyond LLMs (reviewers ryVa, fauP and PYgD)
2. Theoretical analysis is rigorous, with explicit weight constructions and error bounds for each algorithm (reviewer ryVa).
*[The authors wish to emphasize that the constructions are **parameter-efficient**, and the error bounds **contract exponentially with layers**]*
3. Synthetic experiments validate our theory. (reviewers ryVa, BwGA, PYgD)
4. Molecular regression experiment shows that the linear Transformer is capable of learning better positional encodings than standard laplacian eigenvectors on a real-world problem. (reviewers BwGA, PYgD)

## **Weaknesses and Our Response**
> Removing the softmax nonlinearity
> - severely impacts the power of the model (reviewer fauP)
> - limits impact of paper as linear Transformer is not widely used, L2 normalization is also not widely used (reviewer BwGA).

   **Our Response**:
   - Linear attention $Attn^{linear}(Z)$ is a cubic polynomial in $Z$ (**thus not linear in $Z$**). Expressive power of linear Transformer does increase with layers, as seen in all our experiments.
   - There is evidence that **linear Transformers are better suited to the problems considered in this paper**. (For this application, the expressive power of linear Transformer is **not lower** than softmax Transformer.)
   (i) Figures 1 and 2 of [additional plots](https://anonymous.4open.science/r/iclr_2025_rebuttal_plots-3C6/iclr_rebuttal_plots.pdf) show that **linear Transformer outperforms softmax Transformer** in this application.
   (ii) Theory and experiments of [Cheng et al 2023](https://arxiv.org/abs/2312.06528) suggest that softmax activation performs suboptimally compared to linear activation on tasks related to solving linear systems.
   - Appendix A.9 of [von Oswald et al 2023](https://arxiv.org/pdf/2212.07677) suggest that **our constructions for the linear Transformer may be approximated by a multi-head softmax Transformers.**
   - L2 normalization in our paper is similar to **BatchNorm, which outperforms LayerNorm in many graph Transformer experiments.** (see experiments in [(Dwivedi and Bresson 2020)](https://arxiv.org/pdf/2012.09699)).

> Transformers are universal function approximators
> - Theoretical results are not particularly surprising given the use of a linear Transformer (reviewer ryVa)
> - Since the neural network is a universal approximator, it is not surprising that it can solve approximately the hypothesis function space of several graph problems (reviewer PYgD)

   **Our Response**
   - Universal approximation results generally require a very large number of parameters, exponential in dimension or $1/\epsilon$ (see Corollary 4.1, [(Chong 2020)](https://openreview.net/pdf?id=rkevSgrtPr))
   - In contrast, for graphs of $n$ nodes and $m$ edges, our standard construction uses only $O(n^2+m^2)$ parameters per layer. The efficient implementation further reduces this to O(#query^2). Number of layers needed is only $\log(1/\epsilon)$, and sometimes as few as $\log\log(1/\epsilon)$.



> Complexity of the approach is prohibitive, can be $O(n^4)$.

**Our Response**
   - Theoretically, all constructions in our paper can be implemented using the efficient implementation (Section 5), which **requires only O(#query^2) parameters per-layer, independent of the size of the graph**.
   - Experimentally, we show in Figure 3 of [additional plots](https://anonymous.4open.science/r/iclr_2025_rebuttal_plots-3C6/iclr_rebuttal_plots.pdf) that **the efficient implementation performs as-well, and often better-than the standard implementation** on all the problems considered.
   - Table 2 of our paper demonstrates the good performance of the efficient implementation on a real-world molecular ZINC molecular regression problem; **our implementation only used 488 parameters**.

(continued below)

---

> ### Author Response · Authors · 2024-11-22
> **Summary of Reviews and Our Response (cont)**
>
> > Empirical results are limited
> > - performance on ZINC is significantly lower than SOTA (rvYa)
> > - should compare with other predefined PEs such as electric flow or RWPE, as well as other learned PEs (PYgD, BwGA)
>
>    **Our Response**
>    - Despite being worse than SOTA, our LinearTransformer+GraphTransformer implementation significantly outperforms the baseline of LapPE+GraphTransformer (and **LapPE is optimal for GraphTransformer out of numerous fixed PEs** that we tried) . This serves the primary goal of demonstrating that the linear Transformer **can learn a better PE than Laplacian eigenvector in a real-world application**.
>    - Though the baseline uses Laplacian Eigenvectors, the encoding learned by the linear Transformer is **not clearly related to Laplacian eigenvectors**. We conjecture that the learned encoding is a combination of various graph quantities, such as Laplacian eigenvectors, resistive distance, etc, but further analysis is needed.
>    - We acknowledge that more empirical comparisons to different architectures/PEs across different tasks will be useful. However, in many papers the choice of PE and the GNN architecture design are **closely integrated**. Thus properly incorporating the linear Transformer in the architecture to replace the existing PE needs to be done on a case-by-case basis. This is a interesting and valuable problem to pursue, but is beyond the scope of this work.

---

### Meta-Review · Area_Chair_Zu4n · 2024-12-21

**Metareview:**

The paper explores the application of linear Transformers to graph-structured data, demonstrating both theoretical and empirical capabilities in solving canonical graph problems such as electric flow and eigenvector decomposition. The authors provide explicit weight configurations, error bounds, and validate their theoretical claims through synthetic and real-world experiments, particularly focusing on learning positional encodings for molecular regression tasks. The strengths include originality and novelty, theoretical rigor, as well as synthetic experiments consistent with the theory. Suggestion for improvement include conducting more experiments against other PEs and on larger and more diverse datasets, and finding more graph applications for linear Transformers to broaden the impact.

**Additional Comments On Reviewer Discussion:**

The authors have provided comprehensive and constructive responses to the reviewers' concerns. They have clarified theoretical aspects, improved the presentation by adding proof sketches and a conclusion section, and addressed empirical limitations by enhancing their experiments. Notably, Reviewer fauP and Reviewer BwGA acknowledged improvements and adjusted their ratings positively, reflecting the authors' responsiveness and dedication to refining their work.

---

### Decision · Program_Chairs · 2025-01-22

Accept (Poster)